# Associations between NK Cells in Different Immune Organs and Cellular SIV DNA and RNA in Regional HLADR^−^ CD4^+^ T Cells in Chronically SIV_mac239_-Infected, Treatment-Naïve Rhesus Macaques

**DOI:** 10.3390/v14112513

**Published:** 2022-11-13

**Authors:** Xinjie Li, Liyan Zhu, Yue Yin, Xueying Fan, Linting Lv, Yuqi Zhang, Yijin Pan, Yangxuanyu Yan, Hua Liang, Jing Xue, Tao Shen

**Affiliations:** 1Department of Microbiology and Infectious Disease Center, School of Basic Medical Sciences, Peking University, Beijing 100191, China; 2State Key laboratory of Infectious Disease Prevention and Control (SKLID), National Center for AIDS/STD Control and Prevention, Beijing 100206, China; 3Department of Virology, Institute of Laboratory Animal Science, Chinese Academy of Medical Sciences and Peking Union Medical College, Beijing 100021, China

**Keywords:** HIV, SIV, lymph nodes, NK cells, HLADR^−^ CD4^+^ T cells

## Abstract

With the development of NK cell-directed therapeutic strategies, the actual effect of NK cells on the cellular SIV DNA levels of the virus in SIV-infected macaques in vivo remains unclear. In this study, five chronically SIV_mac239_-infected, treatment-naïve rhesus macaques were euthanized, and the blood, spleen, pararectal/paracolonic lymph nodes (PaLNs), and axillary lymph nodes (ALNs) were collected. The distributional, phenotypic, and functional profiles of NK cells were detected by flow cytometry. The highest frequency of NK cells was found in PBMC, followed by the spleen, while only 0~0.5% were found in LNs. Peripheral NK cells also exhibited higher cytotoxic potential (CD56^−^ CD16^+^ NK subsets) and IFN-γ-producing capacity but low PD-1 and Tim-3 levels than those in the spleen and LNs. Our results demonstrated a significant positive correlation between the frequency of NK cells and the ratios of cellular SIV DNA/RNA in HLADR^−^ CD4^+^ T cells (r = 0.6806, *p* < 0.001) in SIV-infected macaques, despite no discrepancies in the cellular SIV DNA or RNA levels that were found among the blood, spleen, and LNs. These findings showed a profile of NK cell frequencies and NK cytotoxicity levels in different immune organs from chronically SIV_mac239_-infected, treatment-naïve rhesus macaques. It was suggested that NK cell frequencies could be closely related to SIV DNA/RNA levels, which could affect the transcriptional activity of SIV proviruses. However, the cytotoxicity effect of NK cells on the latent SIV viral load in LNs could be limited due to the sparse abundance of NK cells in LNs. The development of NK cell-directed treatment approaches aiming for HIV clearance remains challenging.

## 1. Introduction

Cryptic, activatable latent viral reservoirs formed by integrated proviruses remain a major barrier to achieving a functional cure for HIV/SIV infection [1,2,3]. Additionally, lymph node (LN) follicles are considered a major refuge for HIV/SIV viruses due to the role of anatomical reservoirs [4,5,6]. Notably, viral reservoirs are rapidly established and disseminated as early as the acute phase of HIV/SIV infection, even prior to systemic viremia [7,8,9]. Resting CD4^+^ T cells that harbor HIV-1 are characterized by the transcriptional silencing of the provirus and negative expression of surface HLA-II molecules and are, therefore, able to escape from antiretroviral therapy (ART) and even surveillance from the adaptive immune response [10,11].

During resistance to HIV/SIV infection, NK cells mediate multiple immune regulations and effector functions via antibody Fc–FcR interactions, inflammatory factor secretion, and upregulating, activating, or inhibitory receptors [12,13]. Besides harnessing potent natural cytotoxicity and classical antibody-dependent cytotoxicity (ADCC) to eliminate HIV-1 infected cells [14,15], NK cells also induce the apoptosis of target cells by releasing perforin and granzyme promptly [16,17]. In addition, NK cells yield a variety of cytokines and chemokines to prevent the virus from invading CD4^+^ T cells, including but not limited to IFN-γ, TNF-α, and GM-CSF [18,19]. Early events in NK cell rapid proliferation and effector activity are directly involved in the robust control of viral replication in LNs and play a key role in suppressing viremia, as demonstrated in the research of nonpathogenic SIV-infected macaques [20,21,22]. Therefore, multiple NK cell-targeted therapeutic strategies have been explored to achieve a functional cure, aiming to modulate HIV-1 transcription, replication, viral storage, and killing infected cells through inducing activation or enhancing NK cell function [23,24,25].

These HIV/SIV eradication approaches underlie a common philosophy, which is devoted to removing potentially infected lymphocytes and integrating cryptic HIV/SIV DNA from the lymphoid organs. However, due to the heterogeneity in the distribution and immune profile of infiltrated NK cells in different lymphoid organs, the efficacy of cellular SIV DNA levels suppression and viral clearance could be limited. In this study, chronically SIV_mac239_-infected treatment-naive rhesus macaque models were performed to explore the relationship between NK cell distribution, phenotypic and functional characteristics, and the latent SIV DNA load in HLADR^−^ CD4^+^ T cells among blood and secondary lymphoid organs, indicating the potential impacts of NK cell frequencies and functions on the long-term accumulation of cellular SIV DNA and RNA.

## 2. Materials and Methods

### 2.1. Animals and Viral Challenge

Five female Chinese rhesus macaques acquired from and housed at the Institute of Laboratory Animal Science Chinese Academy of Medical Science (ILAS, CAMS) were used in this study. Prior to the experiment, these macaques tested negative for simian type D retrovirus (SRV), simian T lymphotropic viruses (STLV), SIV, cercopithecine herpesvirus 1 (BV), and tubercle bacillus (TB). The animals were aged ranging from 3 to 5 years old and weighed between 3.9 and 4.8 kg at the beginning of the study. The animals were inoculated with 1 mL viral suspension containing 100 TCID_50_ of our SIV_mac239_ challenge stock (a generous gift from P.A. Marx) by the intrarectal route. The basic physiological information, virus inoculation, and euthanasia times for the five macaques were shown in Appendix A. EDTA-anticoagulated blood samples were collected regularly to monitor the dynamic changes in measured CD4^+^/CD8^+^ T cell counts (Appendix A) and SIV viral loads (Appendix A). All assessment experiments of the NK cells from infected rhesus monkeys were performed during the stable phase of SIV chronic infection. After 748-days post-infection, all the macaques were euthanized and necropsied by a professional veterinary pathologist. The blood, spleen, pararectal/paracolonic lymph nodes (PaLNs), and axillary lymph nodes (ALNs) were collected and isolated for subsequent tests and analyses. Additionally, five healthy macaques of matched gender and age were used as negative controls, and their peripheral blood was harvested for peripheral blood mononuclear cells (PBMCs) preparation.

All macaques were cared for in facilities accredited by the Association for Assessment and Accreditation of Laboratory Animal Care (AAALAC) in the Institute of Laboratory Animal Science, Chinese Academy of Medical Sciences (ILAS, CAMS), and under the guidance of an animal laboratory protocol approved by the institutional animal care and use committee and in compliance with animal care procedures. The animals were closely monitored and observed for clinical signs of disease at least twice daily.

### 2.2. Necropsy and Sample Collection

Macaques were euthanized with diazepam (2 mg kg^−1^, SunRise Pharma, Shanghai, China) and ketamine hydrochloride (40 mg kg^−1^, GuTian Pharma, Ningde, China) by intramuscular injection (i.m.). Peripheral blood was collected using EDTA-treated anticoagulation tubes, and PBMCs were isolated by density gradient centrifugation using Histopaque-1077 (Sigma, St Louis, MO, USA) and following the manufacturer’s instructions. Subsequently, each animal was perfused with physiological saline, and a complete necropsy, including the gross and microscopic examination of all tissues and organs, was performed by staff veterinary pathologists at the ILAS. The spleen was minced with a sterile scalpel and ground through a 40 µm cell filter, and erythrocytes were lysed with ACK Lysing Buffer (Gibico). All prepared single-cell suspensions (PBMC, spleen, and LNs) were washed and resuspended in R10 medium (RPMI-1640 supplemented with 10% FBS, 1% Penicillin/Streptomycin, 25 mM HEPES and 2 mM L-Glutamine). A portion of each fresh cell sterile suspension was treated with lysis buffer and cryopreserved for cellular SIV viral RNA and DNA analysis. The remainder was stored at −80 °C in a freezing medium containing 90% fetal bovine serum (FBS, Gibico) and 10% dimethyl sulfoxide (DMSO, Sigma-Aldrich) for further phenotypic and functional analyses.

### 2.3. Plasma and Tissue Viral Load Analyses

The total viral RNA in cell-free plasma was extracted using the viral RNA extraction kit (QIAGEN, Valencia, CA, USA). Extraction and purification procedures were performed according to the manufacturer’s instructions. The RNA purity and protein or TRIzol contamination were quantified through an ND-1000 spectrophotometer (NanoDrop, Wilmington, DE, USA) by measuring A260/280 and A260/230 ratios. The plasma SIV viral load was determined by the 208-bp segment of the SIV_mac239_ gag gene by a quantitative real-time reverse transcription polymerase chain reaction (qRT-PCR) using LightCycler^®^ FastStart DNA Master SYBR Green I (Roche, Indianapolis, IN, USA) and detected using a LightCycler 480 II Real-time PCR Detection System (Roche). The following oligonucleotides were used for real-time RT-PCR: forward primer, 5′-GTAACT ATGTCCACCTGCCATTA-3′ (location: 424–446 bp); reverse primers, 5′-CAGCCTCCTCGTTTATGATGT-3′ (location: 613–633 bp). The RT reaction was carried out with reverse transcriptase by incubation for 15 min at 37 °C and 30 min at 50 °C. The amplification reaction was performed at 95 °C for 10 min for initial activation and followed by 45 cycles at 94 °C for 15 s, 56 °C for 20 s, and 72 °C for 30 s. Standard curves were generated by amplifying the serial dilutions of a reference SIV gag (GI: 530752666). Copy numbers were calculated by interpolation onto the standard curve with the Lightcycler software, version 3.5 [26].

The extraction of SIV DNA from the peripheral blood PBMC was performed using the QIAampDNA Blood Mini Kit (QIAGEN) with 1 × 10^6^ PBMC. Tissue samples, including the spleen and LNs, were processed into single-cell suspensions to complete DNA extraction. The absolute quantification of tissue-specific DNA levels was determined as described previously based on the TaqMan probe technique [8,27,28]. The assay was performed with 100–200 ng sample DNA. SIV DNA standards were prepared by employing the constructed standard plasmid p239SpSp5 (provided by the ILAS, CAMS). The concentration of DNA was detected using NanoDrop and copy numbers which were calculated as a reference linear standard. Forward primer: 5′-GTCTGCGTCATCTGGTGCATTC-3′, reverse primer: 5′-CACTAGGTGTCTCTGCACTATCTGTTTTG-3′, amplification product length: 88 bp; Probe: 5′-(FAM) CTTCCTCAGTGTGTTTCACTTTCTCTTCTGC (BHQ)-3′ (synthesized by Invitrogen). A GAPDH Ready Kit (Invitrogen) was employed as a control. Quantitative RT-PCR was performed in two steps. Step 1: 50 °C for 2 min and 95 °C for 10 min (1 cycle); Step 2: 95 °C for 15 s and 60 °C for 1 min (40 cycles). The assays were accomplished on an ABI StepOne Plus Real-Time PCR System (Applied Biosystems, Foster City, CA, USA).

The total cellular SIV RNA was prepared from 1 × 10^6^ cells using a TRIzol reagent (Invitrogen, Carlsbad, CA, USA), followed by cDNA synthesis using random primers and a SuperScript First-Strand Synthesis System (Invitrogen). The methodology of Taqman real-time PCR for the cellular SIV RNA assay was similar to the above, except that SIV cDNA was used as a template.

### 2.4. NK Cell Stimulation and Functional Assays

The PBMC and spleen were thawed and resuscitated to assess the NK cell function by three stimulation regimens: (1) the NK cell stimulant phorbol myristate acetate (PMA) and ionomycin; (2) co-culture with NK-sensitive K562 cells or (3) CD16 Fc-receptor crosslinking. Specifically, in the PMA plus ionomycin stimuli group, PBMCs and single-cell suspensions were stimulated for 2 h in an R10 culture media containing PMA (100 ng/mL, Sigma-Aldrich) and ionomycin (1 µg/mL, Santa Cruz Biotechnology, Santa Cruz, CA, USA). For the K562 cells co-cultured group, PBMC (or single-cell suspension) and K562 cells were co-incubated at an E: T ratio of 10:1 in the R10 culture media. For CD16 cross-linking, 1 × 10^6^ PBMCs or single-cell suspensions from the spleen were stimulated with 10 μg/mL of the purified anti-CD16 antibody (clone 3G8, Santz Cruz Biotechnology) or mouse IgG1(κ) (clone X40, BD Biosciences) isotype control for 30 min on ice. The cells were washed to remove unbound antibodies and incubated with 10 μg/mL of goat anti-mouse IgG1F(ab’)2 for 5 h (Santa Cruz Biotechnology) at 37 °C. The unstimulated PBMCs (or single-cell suspension) group containing the medium alone was used as a control. All stimulated or unstimulated cells were washed and stained following standard methods, and the expression levels of CD107a and IFN-γ were measured by Flow Cytometry.

### 2.5. Flow Cytometry and Antibodies

PBMCs or single-cell suspension from the spleen and LNs were stained with the following monoclonal antibodies (Abs) for NK cell phenotypic characterization: anti-CD3-Pacific Blue (clone SP34-2, BD Biosciences), anti-CD8APC-Cy7 (clone RPA-T8, BD Biosciences), anti-NKG2A PE (clone Z199, Beckman coulter), anti-CD16 FITC (clone 3G8, BD Biosciences), anti-CD56 PE-Cy5 (clone B159, BD Biosciences), anti-NKp46 Alexa Fluor 700 (29A1.4, BD Biosciences), anti-NKp44PerCP-eFluor 710 (clone 44.189, eBioscience), anti-NKG2D APC (clone 1D11, BD Biosciences), anti-PD-1 PE-eFluor 610 (clone eBioJ105, eBioscience), and anti-Tim-3 Alexa Fluor 700 (clone #344823, R&D).

For simian CD3^+^ CD4^+^ CD8^−^ HLADR^−^ T cells sorting, PBMCs were stained with anti-CD3 V450 (clone SP34-2, BD Biosciences), anti-CD8 APC-Cy7 (clone RPA-T8, BD Biosciences), anti-CD4 PE (clone RPA-T4, BD Biosciences), and anti-HLADR PerCP (clone L243, BD Biosciences). CD3^+^ CD4^+^ CD8^−^ HLADR^−^ T cells were sorted by BD FACS Aria III (BD Biosciences), and only cells with a purity > 95% were used in subsequent experiments.

For the functional characterization of NK cells, anti-CD107a–PE-Cy5 (clone H4A3, BD Biosciences) and 10 μg/mL of Brefeldin-A, 5 μg/mL of Golgi-Stop (BD Biosciences) were added to the PBMCs or single-cell suspensions from the spleen and LNs and were incubated in the R10 culture media for 4 h at 37 °C with 5% CO_2_. The purified cells were first stained with surface markers anti-CD3-Pacific Blue (clone SP34-2, BD Biosciences), anti-CD8APC-Cy7, anti-NKG2A PE, and incubated for 30 min in the dark at room temperature. The cells were then washed with protein-free PBS. The permeabilization step was performed with 1× BD FACS™ Lysing Solution (BD Biosciences), then the cells were washed with protein-free PBS and 0.25% saponin (Sigma-Aldrich). The intracellular IFN-γ of cells were stained with anti-IFN-γ Alexa Fluor 700 (clone B27, BD Biosciences) for 1 h in the dark at room temperature. Then, the cells were washed and fixed in cold 2% paraformaldehyde and transferred to 4 °C for acquisition on a BD LSRII instrument. A minimum of 3 × 10^5^ events were acquired, and all experiments were analyzed using FlowJo 10 software.

### 2.6. Statistical Analyses

Given that the data did not obey a normal distribution, the Kruskal–Wallis test, Mann–Whitney U test, Friedman test, and Spearman’s rank-correlation were applied as appropriate for statistical analyses using GraphPad Prism version 8.0 (GraphPad Software, San Diego, CA, USA) when necessary. Two-tailed *p*-values < 0.05 were considered statistically significant.

## 3. Results

### 3.1. Distributional Characteristics of NK Cells in the Peripheral Blood and Secondary Lymphoid Organs in Chronic SIV_mac239_-Infected Rhesus Macaques

To characterize the distribution of NK cells in different lymphoid organs in the stable stage of chronically SIV_mac239_-infected rhesus macaques, the NK cell frequencies and subpopulations in infected macaques and healthy controls were determined and compared by multicolor flow cytometry. Based on previous studies, the NK cells of rhesus monkeys were identified as CD3^−^CD8α^+^NKG2A^+^ subsets of lymphocytes [29]. The gating strategy of NK cells in PBMCs, PaLNs, ALNs, and spleens is shown in Figure 1A. The cells were gated according to lymphocyte forward and lateral scattering patterns, and after removing dead and adherent cells, CD3^−^CD8α^+^NKG2A^+^ cells were then gated to obtain NK cell subsets. The highest detection rate of NK cells (CD3^−^CD8α^+^NKG2A^+^ cell subsets) was observed in the peripheral blood of healthy macaques, followed by the peripheral blood, spleen, and lymph nodes of SIV chronically infected macaques. Four subsets were further divided based on the expression of NK surface phenotypes, including CD56 and CD16. Specifically, NK cells in PBMC were relatively higher in the healthy controls (median 6.5%, range: 1.1–12.7%) than in the infected macaques (median 1.3%, range: 0.6–4.8%) but did not reach a statistical difference. In SIV-infected macaques, the proportion of NK cells was significantly higher in the PBMC than in the spleen (median 0.42%, range: 0.25–1.7%) or PaLN (median 0.08%, range: 0.03–0.12%) or ALN (median 0.09%, range: 0.05–0.14%). (Multiple comparisons among the infected groups: *p* = 0.001; PBMC vs. Spleen: *p* = 0.05; PBMC vs. PaLN: *p* = 0.008; PBMC vs. ALN: *p* = 0.008; Spleen vs. PaLN: *p* = 0.008; Spleen vs. ALN: *p* = 0.008) (Figure 1B). Further analysis of NK cell subsets revealed a significantly higher percentage of CD56^−^CD16^+^ cell subsets in the peripheral blood and spleen than in the PaLN and ALN. (Multiple comparisons among the infected groups: *p* = 0.001; PBMC vs. PaLN: *p* = 0.008; PBMC vs. ALN: *p* = 0.008; Spleen vs. ALN: *p* = 0.008; PaLN vs. ALN *p* = 0.05) of SIV chronically infected macaques. However, CD56^+^CD16^−^NK cells in healthy macaques’ peripheral blood and ALN of SIV infected macaques were significantly higher than that in the PBMC, spleen, and PaLN of the infected macaques (PBMC vs. healthy control (HC): *p* = 0.05; multiple comparisons among infected groups: *p* = 0.032; PBMC vs. ALN: *p* = 0.032; Spleen vs. ALN: *p* = 0.016; PaLN vs. ALN: *p* = 0.04). No differences were found in the distribution of double negative (DN) cell subsets among the different subgroups (Figure 1C).

### 3.2. NK Cells and CD4^+^ T Cells in Peripheral Blood and Secondary Lymphoid Organs of SIV Chronically Infected Macaques

In SIV-infected macaques, the population of CD4^+^ T cells and HLADR^−^ CD4^+^ T cells provide shelter for SIV DNA storage and amplification; thus, their levels in different organs garnered considerable attention. By assessing CD4^+^ T cells and HLADR^−^ T cells in lymphocyte single-cell suspensions in different lymphoid organs, it was observed that the proportion of CD4^+^ T cells was substantially higher in the peripheral blood than in the spleen and LNs (multiple comparisons: *p* = 0.022; PBMC vs. PaLN: *p* = 0.05; PBMC vs. ALN: *p* = 0.016; PBMC vs. Spleen: *p* = 0.016). Likewise, the proportion of HLADR-T cells in the peripheral blood was significantly greater than in other lymphoid organs (multiple comparisons: *p* = 0.025; PBMC vs. PaLN: *p* = 0.05; PBMC vs. ALN: *p* = 0.016; PBMC vs. Spleen: *p* = 0.032) (Figure 2A). In addition, we further assessed the ratio of NK cells versus CD4^+^ T and HLADR^−^ T cells in different lymphoid tissues of SIV-infected macaques, which to some extent, could reflect the NK cell killing capacity and SIV DNA storage. The ratios of NK to CD4^+^ T cells or NK to HLADR^−^ CD4^+^ T cells were up to and more than 10% in the peripheral blood and the lowest, almost undetectable, in PaLN. The ratio of NK cells to CD4^+^ T cells was considerably higher in the peripheral blood and spleen than in any other lymphoid tissue (multiple comparisons: *p* = 0.001; PBMC vs. PaLN: *p* = 0.008; PBMC vs. ALN: *p* = 0.008; Spleen vs. PaLN: *p* = 0.008; Spleen vs. ALN: *p* = 0.008) (Figure 2B).

### 3.3. Expression of NK Cell Surface Phenotypes in Peripheral Blood and Secondary Lymphoid Organs of SIV Chronically Infected Macaques

NK cell activity and antiviral capacity are extensively influenced by cell-surface activating or inhibitory receptors, which strictly control and regulate the downstream pathways [19]. Therefore, we evaluated the expression of several phenotypical markers of cytotoxicity in healthy and SIV-infected macaques. Given that Tim-3 and programmed death 1 (PD-1) are closely associated with NK cell- and CD8^+^ T cell-mediated cytotoxic suppression [30,31], we first assayed its expression. We found surprisingly that PD-1 and Tim-3 receptor expression was almost absent in healthy macaques, whereas the abundance of PD-1^+^TIM-3^+^ NK cells was dramatically increased in the spleen of the infected macaques (Figure 3A). In particular, PD-1^+^ NK expression levels were highest in the spleen, followed by PaLN, ALN, and PBMC in the infected macaques, while the lowest abundance was found in the PBMC of healthy controls (multiple comparisons among infected groups: *p* = 0.004; Spleen vs. PBMC: *p* = 0.008; Spleen vs. PaLN: *p* = 0.032; Spleen vs. ALN: *p* = 0.008; PBMC vs. HC: *p* = 0.008; PBMC vs. PaLN: *p* = 0.032;). Similarly, frequencies of Tim-3^+^ NK cells in the spleen were far beyond other lymphoid tissues (multiple comparisons among infected groups: *p* = 0.002; Spleen vs. PBMC: *p* = 0.008; Spleen vs. PaLN: *p* = 0.008; Spleen vs. ALN: *p* = 0.008) (Figure 3B). We further observed the expression of NKG2D, NKP44, and NKP46 in the NK cells (Figure 3C). Compared to infected macaques, the highest abundance of NKP46^+^ NK cells (PBMC vs. HC: *p* = 0.008) and the lowest abundance of NKP44^+^ NK cells (PBMC vs. HC: *p* = 0.016) were found in the peripheral blood of the healthy controls. However, among the infected rhesus monkeys, NKP44^+^ NK cell abundance was significantly higher in the spleen compared to other lymphoid organs (multiple comparisons among infected groups: *p* = 0.002; Spleen vs. PBMC: *p* = 0.008; Spleen vs. PaLN: *p* = 0.008; Spleen vs. ALN: *p* = 0.008), while the proportion of NKG2D^+^ NK cells in the PBMC was notably decreased (PBMC vs. HC: *p* = 0.008; multiple comparisons among infected groups: *p* = 0.013; PBMC vs. PaLN: *p* = 0.008; PBMC vs. ALN: *p* = 0.008; PBMC vs. Spleen: *p* = 0.008) (Figure 3D). These findings indicated that, compared to peripheral NK cells, NK cells from secondary lymphoid organs were inclined to an over-activated or exhausted status, which could diminish NK cell-mediated cytotoxicity.

### 3.4. NK Cytotoxic Function Was Impaired in SIV Chronically Infected Macaques

To investigate the cytotoxic capability of NK cells in the spleen and peripheral blood, the expression of CD107a and synthesis of IFN-γ in the lymphocytes from PBMCs and splenic single-cell suspension under three different stimulation conditions (PMA/ionomycin, K562 cells, and CD16 cross-linking) were detected by flow cytometry (Figure 4A). In the presence of PMA and ionomycin stimulants, the expression of CD107a and secretion of IFN-γ in NK cells from both the PBMC and spleen have significantly decreased in SIV chronically infected rhesus monkeys, as compared to the PBMC of healthy rhesus monkeys (PBMC vs. HC: *p* = 0.008, both IFN-γ and CD107a). Additionally, in SIV_mac239_-infected macaques, NK cells in the spleen exhibited weaker cytotoxic capabilities than peripheral NK cells of identical macaques (PBMC vs. Spleen: *p* = 0.024, both IFN-γ and CD107a). The same trend was observed in the K562 cell co-incubation and CD16 cross-linking groups, except for the secretion of IFN-γ in infected macaques between NK cells from PBMCs and the spleen (PBMC vs. HC: *p* = 0.008 (IFN-γ of K562 group); PBMC vs. Spleen: *p* = 0.024 (CD107a of K562 group); *p* = 0.032 (CD107a of CD16 crosslinking group) (Figure 4B). Unfortunately, the cytotoxicity of NK cells in PaLNs and ALNs was not able to be evaluated due to the limited number of NK cells isolated from these LNs. The results above establish that SIV_mac239_-infection leads to impaired NK cytotoxic function and reduced anti-infective and killing capacity.

### 3.5. Positive Correlation between NK Cell Frequency in Lymphocytes and Cellular SIV DNA/RNA Ratio in HLADR^−^ CD4^+^ T Cells

Considering that the ratio of integrated SIV DNA to SIV RNA in the cells of chronically infected macaques could be related to the viral transcriptional activity, the cellular SIV DNA and RNA levels in CD4^+^ and HLADR^−^ CD4^+^ T cells from the blood, spleen, and LNs of untreated infected macaques were assayed. As shown in Table 1, no significant differences in SIV DNA or RNA levels within CD4^+^ T cells and HLADR^−^ CD4^+^ T cells were found among the PBMCs, PaLNs, ALNs, and spleens. However, when the ratios of cellular SIV DNA/RNA were compared, substantially higher DNA/RNA ratios were observed in HLADR^−^ CD4^+^ T cells from PBMCs than that from the PaLNs, ALNs, and spleens (*p* = 0.0055) (Table 1). This result displayed a lower transcriptional activity of SIV DNA in the peripheral blood HLADR^−^ CD4^+^ T cells, indicating a much weaker reactivation status of the latent virus than in other lymphoid tissues and organs.

Furthermore, correlations between the proportion of NK cell subpopulations and SIV DNA/RNA ratios in HLADR^−^ CD4^+^ T cells were analyzed. Interestingly, a positive correlation was found between the frequency of NK cells and the ratios of cellular SIV DNA/RNA in HLADR^−^ CD4^+^ T cells (r = 0.6806, *p* < 0.001). No obvious correlation was found between the cellular SIV DNA/RNA ratio in HLADR^−^ CD4^+^ T cells and the proportion of cytotoxic NK subsets or functional/phenotypic indicators of NK cells (Figure 5). These lines of evidence suggest that the frequencies of NK cells in the lymphocyte tissues and organs could influence the transcriptional activity of the SIV provirus, and a potentially related superior long-term viral control prospect was observed in the peripheral blood compared to any other lymphoid tissue or organ.

## 4. Discussion

In this study, we used an animal model of chronically SIV_mac239_-infected, treatment-naïve rhesus macaques to explore NK cell frequencies and functions in different lymphoid organs and the potential relationship with the cellular SIV DNA and RNA load in regional HLADR^−^ CD4^+^ T cells. NK cell abundance was found to be significantly higher in the peripheral blood, PBMC, and spleen than in any LN, despite a reduction observed when compared to the healthy macaques. Moreover, the lowest frequency of the cytotoxic subset (CD56^−^ CD16^+^ NK cells) was found in LNs. Notably, the proportion of NK/CD4^+^ T cells or NK/HLADR^−^ T cells was also higher in the peripheral blood and spleen than in lymphoid tissues, which hinted at the better control of integrated provirus in the CD4^+^ T cells in the peripheral blood and spleen. In contrast, for LN follicles where the virus was highly aggregated [4], the eradication of infected CD4^+^ T cells posed difficulties due to the relative lack of NK cell abundance.

We further evaluated the phenotypic marker expression and toxic function of NK cells in different lymphoid organs. Herein, PD-1, and Tim-3 exhibited greater levels of peripheral NK cells in chronic SIV_mac239_-infected macaques than in the healthy controls. Furthermore, in infected macaques, NK cells from secondary lymphoid organs, especially in the spleen, expressed higher PD-1, and Tim-3 levels than in the peripheral blood. As biomarkers of activation and the exhaustion/anergy of lymphocytes, higher PD-1 and Tim-3 were highly expressed in NK and T cells and were correlated with the functional impairment of these cells during viral infection [32]. Additionally, higher NKG2D^+^ or NKp44^+^ NK cells were found in the spleen but not in the peripheral blood. The upregulation of these natural cytotoxicity and activation receptors also suggests that NK cells in the spleen were in a state of sustained activation. The assessment of NK cell functions demonstrated that NK cells were functionally impaired following SIV infection, and the potential cytotoxicity of NK cells in the spleen was weaker than that in the peripheral blood. This is consistent with the observation that although sufficient cytotoxic NK cells were present in the spleen in this study, their exhausted phenotypic characteristics resulted in diminished cytotoxic effects. Due to the low frequency of NK cells in the PaLNs and ALNs (usually less than 0.5% of lymphocytes), only a limited number of NK cells could be isolated, which was not sufficient for evaluating the NK function.

In the present study, we found that the ratio of cellular SIV DNA/RNA in the peripheral HLADR^−^ CD4^+^ T cells was higher than that in the PaLNs, ALNs, and spleen in SIV_mac239_-infected, treatment-naïve macaques, suggesting that the SIV RNA transcriptional activity could be more frequent in LNs and the spleen than in the peripheral blood. A previous study found that virus evolution in lymphoid tissue compartments persisted in patients with undetectable virus levels in their peripheral blood [33], indicating that a relatively higher transcriptional level of HIV-1 DNA might be present in the secondary lymphatic tissues. Significantly, the SIV DNA/RNA ratios among different organs were positively correlated with the NK cell frequency in the lymphocytes. It has been evidenced that NK cell frequency inversely correlates with viral load [20,34]; our findings indicate that NK cell frequency could have an impact on the reactivation state of intracellular viruses.

No significant differences were found between the cellular SIV DNA or RNA levels in lymphoid organs from long-term SIV-infected macaques, as neither was an association between the surface phenotypes of NK cells and the magnitude of cellular SIV DNA or RNA load. As shown in Figure 6, unlike in the peripheral blood, the inherent anatomical distance between granular NK cells and HLADR^−^ CD4^+^ T cells in the spleen and LNs make them difficult to approach. Moreover, in the spleen, most NK cells are found in the red pulp and splenic follicle, whereas CD4^+^ T cells are localized primarily in the periarterial lymphatic sheath, where NK cells are absent [35,36]. In addition, the germinal center of LNs is the major distribution area of NK cells, and only a few NK cells are distributed diffusely in the paracortex (T cell-rich) [37]. Accordingly, the influence of NK cells on the cellular SIV viral load of CD4^+^ T cells in the spleen and LNs may be limited.

In recent years, studies have shown that NK cells may clear HIV-1-infected cells through their natural cytotoxicity or ADCC ex vivo or in vitro [38,39]. Cutting-edge research has bloomed, and the development of promising NK cell-directed HIV cure strategies suggests that an improved or modulated immune response could optimize treatment efficacy and enhance the protective immune response against HIV [40,41,42].

## 5. Conclusions

Overall, our findings provide a profile of NK cell frequencies and NK cytotoxicity levels in the peripheral blood, lymphoid tissue, and organs of untreated SIV chronically infected macaques. It was suggested that NK cell frequencies are closely related to SIV DNA/RNA levels, which could affect the transcriptional activity of SIV proviruses. Given the scarcity of NK cells in LNs, the cytotoxicity effect of NK cells on the SIV replication in LNs is likely limited during chronic viral infection. The development of NK cell-directed treatment approaches aimed at HIV clearance remains challenging.

## Figures and Tables

**Figure 1 viruses-14-02513-f001:**
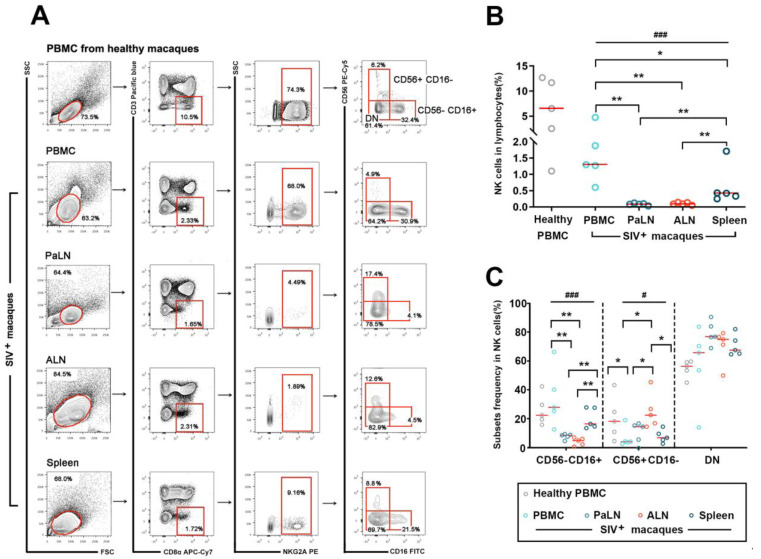
The distribution of simian NK cells and their subsets in peripheral blood and secondary lymphoid organs in SIV-infected macaques. (**A**) Flow cytometric gating defined NK cell subsets (CD3^−^ CD8α^+^ NKG2A^+^) in PBMCs from healthy macaques (*n* = 5) and in single cells of peripheral blood, PaLNs, ALNs, and spleen from SIV-infected macaques (*n* = 5). This illustration shows a representative plot of NK cell subsets form a healthy macaque and an infected macaque. (**B**) The frequencies of NK cells in total lymphocytes from different tissues. (**C**) The distributional characteristics of three NK subsets (CD56^−^ CD16^+^, CD56^+^ CD16^−^, and DN, CD56^−^ CD16^−^). Lines represent median values; little circles in different colors mean five SIV-infected macaques’ percentage levels in different groups. As the data did not obey a normal distribution, after comparing multiple group differences using the Kruskal–Wallis test, the Mann–Whitney U test was performed to compare differences between the two groups. “#” indicates significant differences using the Kruskal–Wallis test (#, *p* < 0.05; ###, *p* < 0.001). “*” indicates significant differences by Mann–Whitney U test (*, *p* < 0.05; **, *p* < 0.01). PBMC—peripheral blood mononuclear cell; PaLN—paracolonic lymph node; ALN—axillary lymph node; DN—double negative.

**Figure 2 viruses-14-02513-f002:**
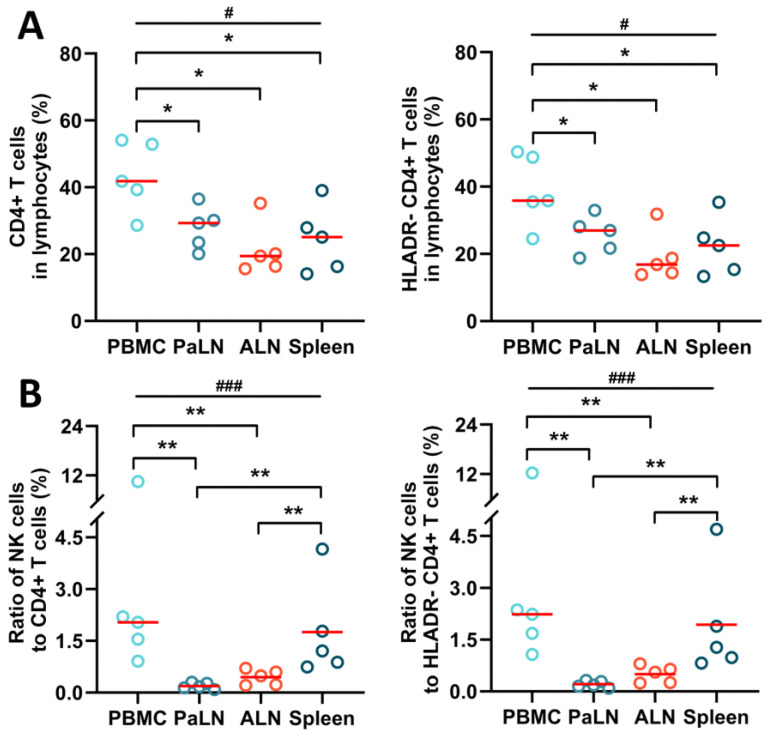
The ratio of NK cells to CD4^+^ T cells and HLADR^−^ CD4^+^ T cells of SIV-infected macaques. (**A**) Levels of CD4^+^ T cells and HLADR^−^ CD4^+^ T cells in single-cell suspension. (**B**) Ratio of NK cells to CD4^+^ T cells and HLADR^−^ CD4^+^ T cells. Lines represent median values; little circles in different colors mean five SIV-infected macaques’ percentage levels in different groups. As the data did not obey a normal distribution, after comparing multiple group differences using the Kruskal–Wallis test, the Mann–Whitney U test was performed to compare differences between the two groups. “#” indicates significant differences using the Kruskal–Wallis test (#, *p* < 0.05; ###, *p* < 0.001). “*” indicates significant differences by Mann–Whitney U test (*, *p* < 0.05; **, *p* < 0.01). PBMC—peripheral blood mononuclear cell; PaLN—paracolonic lymph node; ALN—axillary lymph node.

**Figure 3 viruses-14-02513-f003:**
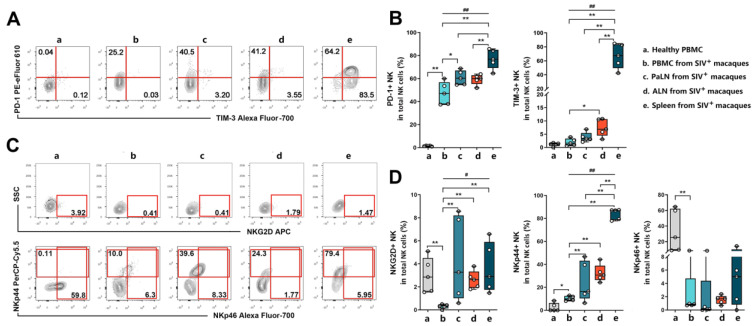
Expression of PD-1, Tim-3, and NK cells receptors on simian NK cells from healthy and SIV-infected macaques. (**A**) and (**C**) are representative cytofluorometric plots for one healthy macaque (a) and SIV-infected macaques (b, c, d, e). Representation of groups a, b, c, d, and e were on the top right corner. Group a was PBMC form healthy macaques; Group b, c, d, e, respectively represented PBMC, PaLN, ALN, spleen from SIV-infected macaques. (**A**) Representative cytofluorometric analysis of PD-1 and Tim-3 expression on simian NK cells. NK cells were gated according to the lymphocyte forward and side scatter pattern and then CD3^−^ CD8α^+^ NKG2A^+^ cells were gated to analyze PD-1 and Tim-3 expression. (**B**) Box-plot analysis for the frequencies of PD-1^+^ and Tim-3^+^NK cells. Lines represent median values, boxes show the 25th and 75th percentiles, and bars show minimum and maximum values. As the data did not obey a normal distribution, after comparing multiple group differences using the Kruskal–Wallis test, the Mann–Whitney U test was performed to compare differences between the two groups. “#” indicates significant differences using the Kruskal–Wallis test (#, *p* < 0.05; ##, *p* < 0.01). “*” indicates significant differences by Mann–Whitney U test (*, *p* < 0.05; **, *p* < 0.01). (**C**) Representative flow cytometric plots indicating expressions of NKG2D, NKp44 and NKp46 on simian NK cells. NK cells were gated according to the lymphocyte forward and side scatter pattern and then CD3^−^ CD8α^+^ NKG2A^+^ cells were gated for analysis. (**D**) Box-plot analysis for the frequencies of NKG2D^+^, NKp44^+^, and NKp46^+^ NK cells. Lines represent median values, boxes show the 25th and 75th percentiles, and bars show minimum and maximum values. As the data did not obey a normal distribution, after comparing multiple group differences using the Kruskal–Wallis test, the Mann–Whitney U test was performed to compare differences between the two groups. “#” indicates significant differences using the Kruskal–Wallis test (#, *p* < 0.05; ##, *p* < 0.01). “*” indicates significant differences by Mann–Whitney U test (*, *p* < 0.05; **, *p* < 0.01). PBMC—peripheral blood mononuclear cell; PaLN—paracolonic lymph node; ALN—axillary lymph node.

**Figure 4 viruses-14-02513-f004:**
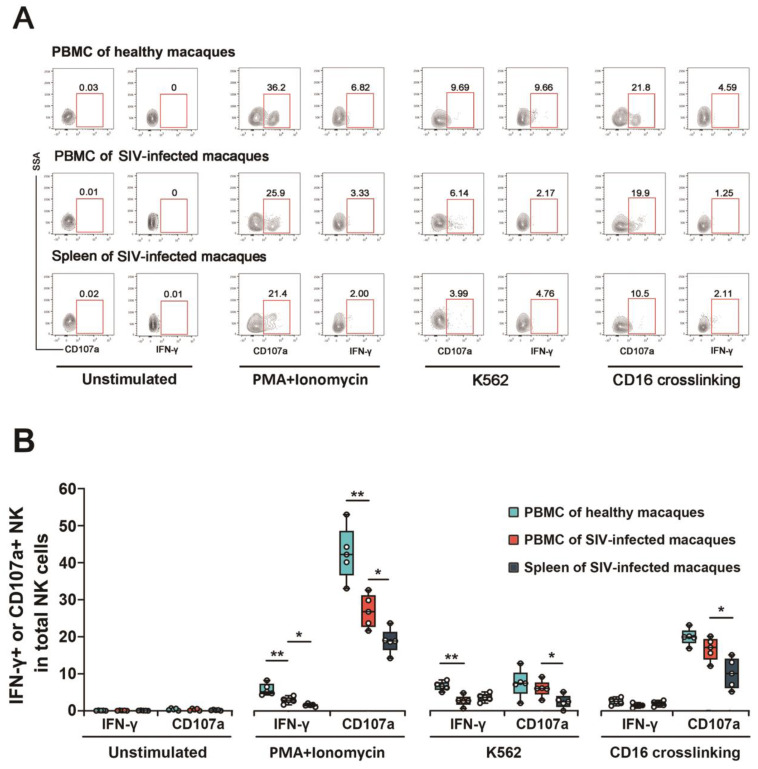
Functional activity of NK cells from simian PBMCs and splenic single-cell suspension. (**A**) Lymphocytes were isolated from the peripheral blood (healthy and SIV-infected macaques) and the splenic single-cell suspension (SIV-infected macaques only) and were stimulated with PMA and ionomycin, K562 cells, and CD16 cross-linking. IFN-γ production and CD107a expression by NK cells were shown. (**B**) The frequencies of IFN-γ^+^ and CD107a^+^ NK cells among total NK cells. “*” indicates significant differences by Mann–Whitney U test (*, *p* < 0.05; **, *p* < 0.01). PBMC—peripheral blood mononuclear cell; PaLN—paracolonic lymph node; ALN—axillary lymph node.

**Figure 5 viruses-14-02513-f005:**
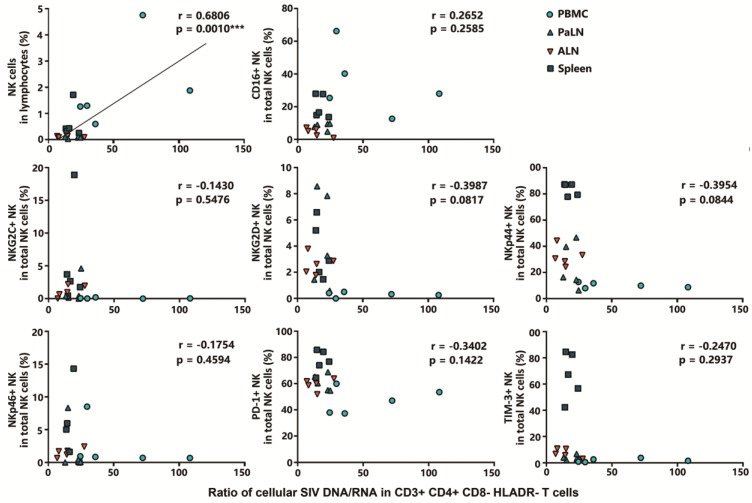
Correlation analysis between frequencies of NK cells in lymphocytes, percentages of cytotoxic NK subsets, phenotypic indicators of NK cells, and cellular SIV DNA/RNA ratios in CD3^+^ CD4^+^ CD8^−^ HLADR^−^ T cells. The correlation was performed using the Spearman rank correlation coefficient test. *p* values < 0.05 were considered a statistical difference (***, *p* < 0.001).

**Figure 6 viruses-14-02513-f006:**
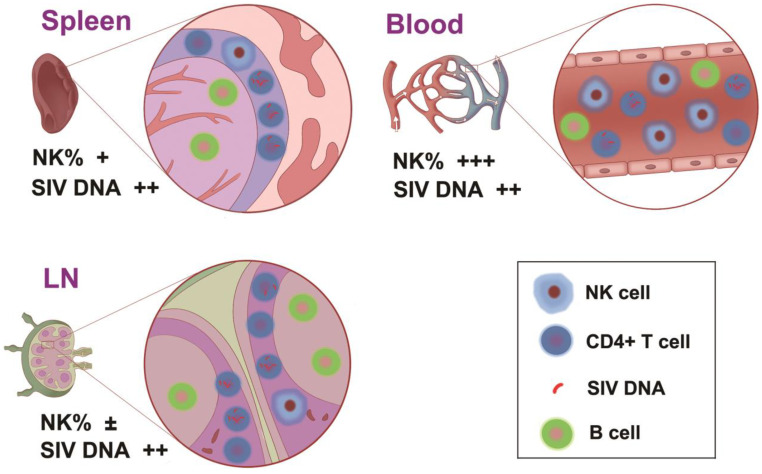
The diagram of NK cells and HLADR^−^ CD4^+^ T cells in the blood, LNs, and spleen.

**Table 1 viruses-14-02513-t001:** Cellular SIV DNA, and RNA levels in CD4^+^ T cells and HLADR^−^ CD4^+^ T cells from different organs and tissues of chronic SIV_mac239_-infected macaques ^1^.

	Animal No.	PBMC	^2^ PaLNs	^3^ ALNs	Spleen	*p* Value
**CD4** ** ^+^ ** **T cells (CD3** ** ^+^ ** **CD4** ** ^+^ ** **CD8^−^)**						
**DNA**	G0101R	440	441	1150	850	0.5206
	G0102R	1554	827	13,666	4908	
	G0104R	3040	1295	861	1201	
	G0105R	2473	4873	7824	1214	
	G0106R	588	2536	1404	851	
**RNA**	G0101R	310	287	455	660	0.1066
	G0102R	73	505	2150	145	
	G0104R	100	242	124	81	
	G0105R	860	632	3237	1049	
	G0106R	81	463	507	313	
**DNA/RNA**	G0101R	1.42	1.54	2.53	1.29	0.2096
	G0102R	21.29	1.64	6.36	3.84	
	G0104R	30.4	5.35	6.94	14.83	
	G0105R	2.87	7.71	2.41	1.18	
	G0106R	7.26	5.48	2.75	2.72	
**HLADR^−^ CD4^+^ T cells (CD3** ** ^+^ ** **CD4** ** ^+^ ** **CD8^−^HLADR^−^)**						
**DNA**	G0101R	3971	1100	3693	4550	0.5206
	G0102R	33,338	2828	14,670	10,199	
	G0104R	6391	4474	927	1795	
	G0105R	2863	12791	9944	1689	
	G0106R	2594	6322	2185	1335	
**RNA**	G0101R	163	72	257	323	0.7709
	G0102R	463	216	2089	424	
	G0104R	178	194	112	122	
	G0105R	97	553	359	102	
	G0106R	24	254	146	68	
**DNA/RNA**	G0101R	24.36	15.28	14.37	14.08	0.0055 **
	G0102R	72	13.09	7.02	24.05	
	G0104R	35.9	23.06	8.27	14.71	
	G0105R	29.51	23.13	27.7	16.56	
	G0106R	108.08	24.89	14.96	19.63	

^1^ All tissue values are the means for two different tissue samples. All viral load values are normalized to 1 × 10^6^ cells in each sample. ^2^ PaLNs—pararectal/paracolonic lymph nodes. ^3^ ALNs—axillary lymph nodes. The Friedman test was used to statistically analyze the overall distribution of the paired multiple groups (differences across PBMC, PaLNs, ALNs, and Spleen groups) because the data were matched and did not follow a normal distribution. *p* values < 0.05 were considered a statistical difference (**, *p* < 0.01).

## Data Availability

The datasets used and/or analyzed during the current study are available from the corresponding author on reasonable request.

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
