# Peer review of "Associations between NK Cells in Different Immune Organs and Cellular SIV DNA and RNA in Regional HLADR CD4+ T Cells in Chronically SIVmac239-Infected, Treatment-Naïve Rhesus Macaques"

_viruses, 2022, doi:10.3390/v14112513_

Round 1

Reviewer 1 Report

The aim of this work was to study the role of NK cells in the control of SIV reservoir in resting CD4+ T cells from blood and lymphoid tissues (spleen and lymph nodes). The authors showed an association between NK frequency among lymphocytes and a decreased SIV reservoir transcriptional activity (cell-associated SIV RNA) reported to the SIV reservoir size (SIV DNA). Non-human primate model is essential to better understand pathogenesis of HIV infection in human. In this particular context, this kind of study is valuable to progress in the field of in NK cell-targeted therapeutic strategies for HIV functional cure.

General concept comments

Despite interesting topic, in some places, methods and results meet limitations and need to be presented in a more accurate way. As an example, virologic parameters used in this study to characterize SIV reservoir do not allow to assess “integrated SIV DNA” or “replication” per se. In this study, there is no quantification of proviral DNA (ie integrated DNA) as announced, but quantification of total SIV DNA (including free linear or episomal DNA). In the same manner, intracellular RNA only allows to appreciate transcription but not replication. Biologic interpretation from these parameters should be re-thought accordingly and should not lead to excessive assumptions. More generally, the study lacks detail in places and need clarification in particular considering the presentation of results and figures. Finally, throughout the manuscript, some language errors could be checked.

Specific comments

Abstract:

L39: “Peripheral NK cells also exhibited higher cytotoxic potential ». The authors should precise by which parameter the cytotoxic potential is assessed. If it is by frequency of CD56- CD16+ NK, cytotoxic potential is higher compared with that of LN but not spleen (see Figure 1C), if it is by the CD107a expression, in that case, cytotoxic potential is higher compared with that of spleen (no data available for LN) (see Figure 4). The authors should clarify this point.

L42 Statistical analyse result indicated into bracket (r = 0.6813, P < 0.001) does not correspond to that indicated in Figure 5 (r= 0.6806). The authors should clarify this point.

L43 SIV DNA is always intracellular. « intracellular » SIV DNA is a pleonasm. The authors should change accordingly throughout the manuscript. Noteworthy, by contract it is very important to precise « intracellular » for SIV RNA.

L45 « reactivation state of intracellular viruses » should be changed by « transcriptional activity of proviruses ».

Materials and methods

Figure S1. The authors should add in legend « in bloodstream ».

L125 « SIV viral RNA and proviral DNA analysis ». Same remark: authors should precise intracellular for RNA and remove proviral.

L134 To my mind, SYBR Green I Master Mix does not contain a reverse transcriptase… May the authors explain how the prior reverse transcription is performed (enzyme, primer (hexamer or specific) ?

L136-137-151-153 May the authors add coordinates of primers ?

L143 « proviral » should be removed

L168-171-177 « PBMCs » but also single-cell suspension I guess?

L189 A comma is needed after “sorting”.

L194 The authors should add “anti-“ before “CD107a–phycoerythrin-Cy5”. Throughout the paragraph, authors should choose between phycoerythrin or PE.

Results

L214 First paragraph entitled “Distributional characteristics of NK cells in the peripheral blood and secondary lymphoid organs in chronic SIVmac239-infected rhesus macaques” and Figure 1. The authors should indicate in the figure 1A, the lymphocytes on the FSC x SSC gate and add in the legend that it is a representative dot plot for one healthy animal and one infected animal. The authors should remove the « s » of macaques. The selection of NK cells based on CD3- CD8alpha+ NKG2A+ should be added in the manuscript not only specified in the legend.

L239 Please expand « DN » => double negative

L241 Second paragraph entitled “NK cells and CD4+ T cells in peripheral blood and secondary lymphoid organs of SIV chronically infected macaques” and Figure 2. The authors should precise the cell set from which the frequency of NK cells is appreciated. Sometimes it is in lymphocytes L246, then in cells in peripheral blood L249. In Figure 2A, should be added % of the cell set in question; lymphocytes, I suppose. L339 In the legend of figure 2B it is written Ratio of NK cells in CD4+ T cells??... I don’t think that the proportion of NK cells is assessed among CD4+ T cells. Please be careful. Authors should reformulate as well title of y-axis of graphs. Additionally, the fact that the 2 graphs of Figure 2B are exactly the same sounds odd. The authors have to check if these graphs are the right ones. Finally, the authors should add in this section the results obtained from PBMCs of healthy macaques and be careful to change the title of this paragraph accordingly.

L 259 Third paragraph entitled “Expression of NK cell surface phenotypes in peripheral blood and secondary lymphoid organs of SIV chronically infected macaques” and Figure 3. The authors should add in the legend of Figure 3 that Figure 3A and 3B are a representative dot plot for one healthy animal (a) l and one infected animal (b,c,d,e). Besides, no significant difference is represented in Figure 3C between PBMCs of healthy macaques and lymphoid organs of infected macaques considering the expresion of TIM3+.  It is not in accordance with « We found surprisingly that PD-1 and Tim-3 receptor expression was almost absent in healthy macaques whereas the abundance of PD-1+TIM-3+ NK cells was dramatically increased in various lymphoid organs of infected macaques » L266-268. The authors should clarify this point. Figure3A, B, C do not appear in the text in the alphabetic order. The authors are kindly requested to modify this order accordingly.

L287 Fourth paragraph entitled “NK cytotoxic function was impaired in SIV chronically infected macaques” and Figure 4. L289 « expression of CD107a and synthesis of IFN-γ in lymphocytes » is assessed in lymphocytes (as it is written) or NK cells? Instead of « from splenic lymphocytes » I would write « from splenic single-cell suspension ».

L291-295 It is not clear if it is in NK cells. The authors should reformulate and be more specific.

L294 Please expand « HC ».

L297 « The same trend was observed in the K562 cell co-incubation and CD16 cross-linking groups same trend » except for secretion of IFN-gamma in infected macaques between NK cells from PBMCs and spleen if I consider the Figure 4B. Authors should clarify this point.

L359 Legend of Figure 4: Is it “lymphocytes” or “PBMCs and single-cell suspension” as it is described in the method section?

L304 Fifth paragraph entitled “Positive correlation between NK cell frequency and intracellular SIV DNA/RNA ratio in resting CD4+ T cells” and Figure 5 and Table 1. In this subtitle and in the y-axis of the first graph of Figure 5, the authors should precise the cell set from which the frequency of NK cells is appreciated.

L307 « suggests » is not suitable here. The authors should reformulate.

L317 It would be interesting to have the results in CD4+ T cells as well, not only in resting cells.

L318-320 Does the positive correlation remain when examining lymphoid sites separately? L323 « in the lymphocyte tissues and organs ». Do the authors mean «lymphoid »?

L324 « Transcriptional activity » would be better than « reactivation status », « SIV provirus » would be better than latent SIV integrated virus

L325 I would write « a potentially related superior long-term viral control prospect », « was observed » better than «was demonstrated».

L365 In Table 1 legend « Cellular SIV DNA and RNA levels in single-cell suspension and resting CD4 T cells from different tissues of chronic SIVmac239-infected macaques », « in CD4 T cells » instead of « in single-cell suspension » would be better as it is represented in the table.

Discussion and figure 6

L385 The authors should define « cytotoxic subset »

L398 « Additionally, higher NKG2D+ or NKp46+ NK cells were found in the spleen but not in the peripheral blood ». Figure 3 doesn’t show this result for NKp46+ NK cells. The authors should clarify this point accordingly.

L409 Authors should define «non-cytotoxic subset»

L412-423 Hypothesis should be rephrased with more restraint given that cell-associated RNA assesses transcription not replication.

L424 « No significant differences were found between intracellular SIV DNA or RNA levels in lymphoid organs » AND in PBMCs (see Table 1). Anatomical distance explanation should have been proposed to explain difference between PBMCs and lymphoid organs considering the ratio DNA/RNA, not considering DNA and intracellular RNA levels separately since there is no significant difference between sites for these two biomarkers. This paragraph is quite confusing. The authors should clarify it.

I think that the Figure 6 should represent transcriptional activity not just SIV DNA as the main message of this article is about the ratio DNA/intracellular RNA.

L457 all instead of All

Author Response

Point-by-point responses to the comments from the Editors and Reviewers

Reviewer 1

The aim of this work was to study the role of NK cells in the control of SIV reservoir in resting CD4+ T cells from blood and lymphoid tissues (spleen and lymph nodes). The authors showed an association between NK frequency among lymphocytes and a decreased SIV reservoir transcriptional activity (cell-associated SIV RNA) reported to the SIV reservoir size (SIV DNA). Non-human primate model is essential to better understand pathogenesis of HIV infection in human. In this particular context, this kind of study is valuable to progress in the field of in NK cell-targeted therapeutic strategies for HIV functional cure.

General concept comments

Despite interesting topic, in some places, methods and results meet limitations and need to be presented in a more accurate way. As an example, virologic parameters used in this study to characterize SIV reservoir do not allow to assess “integrated SIV DNA” or “replication” per se. In this study, there is no quantification of proviral DNA (ie integrated DNA) as announced, but quantification of total SIV DNA (including free linear or episomal DNA). In the same manner, intracellular RNA only allows to appreciate transcription but not replication. Biologic interpretation from these parameters should be re-thought accordingly and should not lead to excessive assumptions. More generally, the study lacks detail in places and need clarification in particular considering the presentation of results and figures. Finally, throughout the manuscript, some language errors could be checked.

Specific comments

Abstract:

  1. L39: “Peripheral NK cells also exhibited higher cytotoxic potential ». The authors should precise by which parameter the cytotoxic potential is assessed. If it is by frequency of CD56- CD16+ NK, cytotoxic potential is higher compared with that of LN but not spleen (see Figure 1C), if it is by the CD107a expression, in that case, cytotoxic potential is higher compared with that of spleen (no data available for LN) (see Figure 4). The authors should clarify this point.

A: Thank you very much for pointing this out. As shown in Figure 1, the cytotoxic potential (median level) represented by CD56- CD16+ subset of peripheral NK cells is higher than that of LN and spleen (please see Figure 1C). Given the absence of data for the LN in the assessment of cytotoxic potential through CD107a expression (see Figure 4), we have redescribed the term "cytotoxic potential" as "CD56- CD16+ NK" cell proportion in the abstract to make the statement more clarifying. Please see line 37 in the revised manuscript.

  1. L42 Statistical analyse result indicated into bracket (r = 0.6813, P < 0.001) does not correspond to that indicated in Figure 5 (r= 0.6806). The authors should clarify this point.

A: Thank you very much for your careful review. This was a clerical error and we have corrected the statistical analysis here to r=0.6806. Please see line 39 in the revised manuscript.

  1. L43 SIV DNA is always intracellular. « intracellular » SIV DNA is a pleonasm. The authors should change accordingly throughout the manuscript. Noteworthy, by contract it is very important to precise « intracellular » for SIV RNA.

A: We appreciate your suggestion. Considering that DNA and RNA levels in cells are assessed and evaluated simultaneously, we have changed the term "intracellular" to "cellular" throughout to avoid redundancy and to limit the definition of RNA. We hope this will make the statement more accurate. Please see lines 31-40 in the revised manuscript for example.

  1. L45 « reactivation state of intracellular viruses » should be changed by « transcriptional activity of proviruses ».

A: Thank you for bringing this to our attention. We have updated the language to reflect more accurate terminology. Instead of "reactivation state of intracellular viruses," we have changed it to "transcriptional activity of proviruses." This can be seen on line 44 of the revised manuscript.

Materials and methods

  1. Figure S1. The authors should add in legend « in bloodstream ».

A: We have added "in the bloodstream" to the new legend. Please see the line 15 of revised Figure S1 in the supplementary materials for details.

  1. L125 « SIV viral RNA and proviral DNA analysis ». Same remark: authors should precise intracellular for RNA and remove proviral.

A: Thank you for your advice regarding the removal of the term "proviral" and the clarification of intracellular RNA. We have made the changes you suggested throughout the text, as mentioned in Question 3. Please see line 122 in the revised manuscript.

  1. L134 To my mind, SYBR Green I Master Mix does not contain a reverse transcriptase… May the authors explain how the prior reverse transcription is performed (enzyme, primer (hexamer or specific)?

A: Thank you for bringing this to our attention. In the previous method, we described the reverse transcription procedure and the amplification reaction procedure jointly. However, we realize that this was inadequate and have therefore added the detailed steps and methods of reverse transcription in the new manuscript. Please see lines 131-140 of the revised manuscript for more information.

  1. L136-137-151-153 May the authors add coordinates of primers ?

A: Thank you for the reminder. We have added coordinates for the primers. Please see lines 135-137 in the revised manuscript.

  1. L143 « proviral » should be removed

A: Thank you for your helpful suggestion. We have removed the "proviral" from the text. Please see lines 144-147 of the revised manuscript for more details.

  1. L168-171-177 « PBMCs » but also single-cell suspension I guess?

A: Thank you for your insightful feedback. It is indeed true that single-cell suspension of spleen should be included in this study. We have added this to the revised manuscript and are grateful for the reminder. Please see lines 169-178 in the revised manuscript.

  1. L189 A comma is needed after “sorting”.

A: Thank you for your attentive review. We have added a comma after "sorting" in the revised manuscript. Please see line 191 in the revised manuscript.

  1. L194 The authors should add “anti-“ before “CD107a–phycoerythrin-Cy5”. Throughout the paragraph, authors should choose between phycoerythrin or PE.

A: Thank you very much for your kind reminder. With this suggestion, we have added "anti-" before CD107a–PE-Cy5, and replaced “phycoerythrin” with “PE”. Please see line 19 6 in the revised manuscript.

Results

  1. L214 First paragraph entitled “Distributional characteristics of NK cells in the peripheral blood and secondary lymphoid organs in chronic SIVmac239-infected rhesus macaques” and Figure 1. The authors should indicate in the figure 1A, the lymphocytes on the FSC x SSC gate and add in the legend that it is a representative dot plot for one healthy animal and one infected animal. The authors should remove the « s » of macaques. The selection of NK cells based on CD3- CD8alpha+ NKG2A+ should be added in the manuscript not only specified in the legend.

A: Thank you very much for the advice. In accordance with the guidance, we have provided additional descriptions in the figure legend and illustrated the NK cell selection strategy in the manuscript. Please see lines 221-225 and lines 335-343 in the revised manuscript.

  1. L239 Please expand « DN » => double negative

A: Thank you very much for your suggestion. We have added the description of “DN” in the revised manuscript. Please see line 243 in the revised manuscript.

  1. L241 Second paragraph entitled “NK cells and CD4+ T cells in peripheral blood and secondary lymphoid organs of SIV chronically infected macaques” and Figure 2. The authors should precise the cell set from which the frequency of NK cells is appreciated. Sometimes it is in lymphocytes L246, then in cells in peripheral blood L249. In Figure 2A, should be added % of the cell set in question; lymphocytes, I suppose. L339 In the legend of figure 2B it is written Ratio of NK cells in CD4+ T cells??... I don’t think that the proportion of NK cells is assessed among CD4+ T cells. Please be careful. Authors should reformulate as well title of y-axis of graphs. Additionally, the fact that the 2 graphs of Figure 2B are exactly the same sounds odd. The authors have to check if these graphs are the right ones. Finally, the authors should add in this section the results obtained from PBMCs of healthy macaques and be careful to change the title of this paragraph accordingly.

A: Thank you very much for your instructive advice. 1) You are absolutely right that we should clarify the source of NK cells and CD4+ T cells. Realizing this, we have illustrated the sources of all cells in the Figure 1 to Figure 5 (please see new Figure 1 to Figure 5) and have modified the legends and manuscript accordingly. 2) In fact, Figure 2B has compared the proportion of NK cells to the proportion of CD4+ T cells and calculated the ratios, and for clarification, we have modified Figure 2B and the legend accordingly (please see new Figure 2 and the figure legend and line 258). 3) Last but not least, we have double-checked the data of Figure 2B. Although there are similarities between the left and right panel values in Figures 2A and 2B, there are actually differences between the two sets of data. For example, the highest value in the PBMC group on the left panel of Figure 2B is 10.52, whereas this highest value in Figure 2B is 12.29. There are differences in the other values as well, but there may be similarities due to the proportions of cells presented. We can provide the raw data from this section for your reference if required for better clarification.

  1. L259 Third paragraph entitled “Expression of NK cell surface phenotypes in peripheral blood and secondary lymphoid organs of SIV chronically infected macaques” and Figure 3. The authors should add in the legend of Figure 3 that Figure 3A and 3B are a representative dot plot for one healthy animal (a) l and one infected animal (b,c,d,e). Besides, no significant difference is represented in Figure 3C between PBMCs of healthy macaques and lymphoid organs of infected macaques considering the expresion of TIM3+. It is not in accordance with « We found surprisingly that PD-1 and Tim-3 receptor expression was almost absent in healthy macaques whereas the abundance of PD-1+TIM-3+ NK cells was dramatically increased in various lymphoid organs of infected macaques » L266-268. The authors should clarify this point. Figure3A, B, C do not appear in the text in the alphabetic order. The authors are kindly requested to modify this order accordingly.

A: Thank you for pointing out these problems. 1) Aware of this shortcoming, we have added to the revised Figure 3 legend that those are representative plots of a healthy animal (a) and an infected animal (b, c, d, e). Please see line 355 in the revised manuscript. 2) In Figure 3C (the new Figure 3B), there was indeed a difference between PBMCs of healthy macaques and lymphoid organs of infected macaques in TIM3+ expression. However, given the fact that the PBMCs of healthy macaques cannot be directly compared to the lymphoid organs from infected macaques, the P-values between these two are not shown in the figure. We could see visually in the new Figure 3B that NK cells in PBMCs from healthy macaques lack TIM-3+ receptor expression, whereas NK cells in PBMCs from infected macaques have elevated TIM-3+ receptor expression, and that expression of other lymphoid organs from infected macaques is higher than PBMCs from infected macaques, leading us to this description. The difference in the representative cytofluorometric plot on the left is not obvious because it is only one of the samples. For clarity, we have replaced "various lymphoid organs" with "spleen" in the revised text. Please see new line 268 in the revised manuscript. 3) With regard to your suggestion that figures 3A, B and C do not appear in alphabetical order in the text, we have promptly revised this and ensured that the descriptions and letters are in order. Please see new Figure 3 and lines 273-287 in the revised manuscript.

  1. L287 Fourth paragraph entitled “NK cytotoxic function was impaired in SIV chronically infected macaques” and Figure 4. L289 « expression of CD107a and synthesis of IFN-γ in lymphocytes » is assessed in lymphocytes (as it is written) or NK cells? Instead of « from splenic lymphocytes » I would write « from splenic single-cell suspension ».

A: Thank you very much for mentioning this. We have assayed the cytotoxic function of NK cells by conducting experiments with lymphocytes (analyzing the NK population in lymphocytes through flow cytometry gates). As the circle gate process for NK cells has been described in the previous sections, it is not repeated here. And following your suggestion, we have replaced "from splenic lymphocytes" with "from splenic single-cell suspension". Please see line 293 in the revised manuscript.

  1. L291-295 It is not clear if it is in NK cells. The authors should reformulate and be more specific.

A: Thank you for your advice. To clarify that this assessment is in NK cells, we have reorganized the sentence. Please see lines 296-298 in the revised manuscript.

  1. L294 Please expand « HC ».

A: Thank you for your patience in reviewing this. We have added a description of "HC" to make the content more complete. Please see line 241 in the revised manuscript.

  1. L297 « The same trend was observed in the K562 cell co-incubation and CD16 cross-linking groups same trend » except for secretion of IFN-gamma in infected macaques between NK cells from PBMCs and spleen if I consider the Figure 4B. Authors should clarify this point.

A: Thank you very much for pointing this out. In order to clarify the description and make the content more accurate, we have added the relevant descriptions in the revised manuscript. Please see lines 303-304 in the revised manuscript.

  1. L359 Legend of Figure 4: Is it “lymphocytes” or “PBMCs and single-cell suspension” as it is described in the method section?

A: Thank you for your suggestion. As described in the methods, "lymphocytes" from peripheral blood and splenic single-cell suspension were isolated for the experiment, and the function of the NK cell subsets was tested by flow cytometry. As the flow assay method has been mentioned before, it has not been repeated here. For clarity of meaning, we have added a description of the content in the figure legend. Please see lines 372-377 in the revised manuscript.

  1. L304 Fifth paragraph entitled “Positive correlation between NK cell frequency and intracellular SIV DNA/RNA ratio in resting CD4+ T cells” and Figure 5 and Table 1. In this subtitle and in the y-axis of the first graph of Figure 5, the authors should precise the cell set from which the frequency of NK cells is appreciated.

A: Thank you for the reminder. Based on this suggestion, we have revised the description of the source of NK cell frequencies in Figure 5 and the subtitle. Please see lines 310-311, 387-390 and the new Figure 5 in the revised manuscript.

  1. L307 « suggests » is not suitable here. The authors should reformulate.

A: Thank you for your advice. We realized that the expression of the sentence did not make sense and have replaced the term "suggests" with a more appropriate word. Please see line 313 in the revised manuscript.

  1. L317 It would be interesting to have the results in CD4+ T cells as well, not only in resting cells.

A: Thank you for your nice suggestion. However, we did not find a significant difference among total CD4+ T cells and therefore the data were not shown.

  1. L318-320 Does the positive correlation remain when examining lymphoid sites separately? L323 « in the lymphocyte tissues and organs ». Do the authors mean «lymphoid »?

A: Thank you for highlighting this. Our previous attempts to examine lymphoid sites separately were unsuccessful due to the limited sample size of the data. This prevented us from coming up with a correlation comparison. As a result, we decided to present the overall results instead. It's worth noting that "in the lymphocyte tissues and organs" refers to lymph nodes and spleens.

  1. L324 « Transcriptional activity » would be better than « reactivation status », « SIV provirus » would be better than latent SIV integrated virus

A: We appreciate the reminder and have made the suggested changes. "Reactivation status" has been replaced with "transcriptional activity" and "latent SIV integrated virus" is now "SIV provirus". These changes can be seen on line 330 of the revised manuscript.

  1. L325 I would write « a potentially related superior long-term viral control prospect », « was observed » better than «was demonstrated».

Thank you for your insightful advice. The sentence has been rewritten in accordance with your suggestions for improved clarity. Please see line 331 in the revised manuscript.

  1. L365 In Table 1 legend « Cellular SIV DNA and RNA levels in single-cell suspension and resting CD4+ T cells from different tissues of chronic SIVmac239-infected macaques », « in CD4+ T cells » instead of « in single-cell suspension » would be better as it is represented in the table.

A: Thank you for your suggestion. We have changed the term "single-cell suspension" to "CD+ T cells" to make the denotation clearer. Many thanks to your careful reminder. Please see line 379 in the revised manuscript.

Discussion and figure 6

  1. L385 The authors should define « cytotoxic subset »

A: Thank you for your helpful reminder. Aware of this, we have promptly added to the meaning of "cytotoxic subset". Please see lines 402-403 in the revised manuscript.

  1. L398 « Additionally, higher NKG2D+ or NKp46+ NK cells were found in the spleen but not in the peripheral blood ». Figure 3 doesn’t show this result for NKp46+ NK cells. The authors should clarify this point accordingly.

A: Thank you very much for pointing this out. We apologize for the typo, "NKp46+ " here should be rectified to "NKp44+". In fact, we intended to mean that higher NKG2D+ or NKp44+ NK cells were found in the spleen but not in the peripheral blood. We have corrected this in the manuscript. Please see line 415 in the revised manuscript. Thank you for your careful examination.

  1. L409 Authors should define «non-cytotoxic subset»

A: Thank you for the reminder. We realize that "non-cytotoxic subset" is an inaccurate statement and have therefore amended this description in the revised manuscript. Please see line 426 in the revised manuscript.

  1. L412-423 Hypothesis should be rephrased with more restraint given that cell-associated RNA assesses transcription not replication.

A: Your suggestion brings greater clarity to the matter. It was premature of us to make such a hasty assumption. In order to make the statement more objective and reasoned, we have rewritten the sentence to limit the content to "transcription". Please see lines 431-432 in the revised manuscript.

  1. L424 « No significant differences were found between intracellular SIV DNA or RNA levels in lymphoid organs » AND in PBMCs (see Table 1). Anatomical distance explanation should have been proposed to explain difference between PBMCs and lymphoid organs considering the ratio DNA/RNA, not considering DNA and intracellular RNA levels separately since there is no significant difference between sites for these two biomarkers. This paragraph is quite confusing. The authors should clarify it.I think that the Figure 6 should represent transcriptional activity not just SIV DNA as the main message of this article is about the ratio DNA/intracellular RNA.

A: Your suggestions are very instructive. Since our results only reflect differences in DNA/RNA levels, it is not appropriate to discuss DNA or RNA levels separately. Therefore, we removed the discussion of DNA and RNA loadings from this paragraph and only presented the interpretation of anatomical distances. Please see lines 443-450 in the revised manuscript.

  1. L457 all instead of All

A: Thank you for your careful inspection. We have changed "All" to "all". Please see line 472 in the revised manuscript.

Reviewer 2 Report

In this manuscript, Li et al evaluated NK cell activity and the relationship between NK cells and SIV DNA and RNA levels in scenario of SIV chronical infection in macaques. Results showed that peripheral NK cells may possess the highest abundance and cytotoxic activities compared with that from other lymphoid organs. In addition, the frequency of NK cells and the intracellular SIV DNA/RNA ratio showed positive correlation, which may depict the potential effect of NK cells on SIV reactivation. Overall, this is a well-designed study with interesting and significant conclusions. However, some major points concerning the figure display and data interpretation must be clarified.

1 For the flow plots in Fig 1A and Fig 3A-B, please add Y/X-axis values as shown in Fig 4A.

2 The left and right panels in Fig 2A and 2B showed extreme similarity.

3 In lines 253-254, how to explain “…which to some extent could reflect the NK cell killing capacity and SIV DNA storage…”? If this conclusion came from former publications, please cite the references.

4 How did you calculate the P values in Table 1?

5 Please explain why most of the statistics in figures had identical P values (multiples of 0.008)? For example, most two asterisks signified P=0.008, while one asterisk signified P=0.016, 0.024 or 0.032.

Author Response

Point-by-point responses to the comments from the Editors and Reviewers

Reviewer 2

Comments and Suggestions for Authors

In this manuscript, Li et al evaluated NK cell activity and the relationship between NK cells and SIV DNA and RNA levels in scenario of SIV chronical infection in macaques. Results showed that peripheral NK cells may possess the highest abundance and cytotoxic activities compared with that from other lymphoid organs. In addition, the frequency of NK cells and the intracellular SIV DNA/RNA ratio showed positive correlation, which may depict the potential effect of NK cells on SIV reactivation. Overall, this is a well-designed study with interesting and significant conclusions. However, some major points concerning the figure display and data interpretation must be clarified.

  1. For the flow plots in Fig 1A and Fig 3A-B, please add Y/X-axis values as shown in Fig 4A.

A: Thank you very much for your instructive advice. Based on this comment, we have modified the flow charts in Figures 1 and 3 by adding values for the Y/X axis as in Figure 4A. Please see new Figure 1 and Figure 3 in the revised manuscript.

  1. The left and right panels in Fig 2A and 2B showed extreme similarity.

Thank you for pointing this out. We have verified the data and there are indeed discrepancies between the left and right panel values in Figures 2A and 2B. For example, the highest value in the PBMC group on the left panel of Figure 2B is 10.52, while the corresponding value on the right panel is 12.29. Although there are similarities between some of the values, this is likely due to the proportions of cells presented. In our detection results, HLADR-CD4 +T cells accounted for about 85-90% of the total CD4+T cells. We would be happy to provide the raw data from this section for your reference if required for better clarification.

  1. In lines 253-254, how to explain “…which to some extent could reflect the NK cell killing capacity and SIV DNA storage…”? If this conclusion came from former publications, please cite the references.

A: Thank you very much for your suggestion. In this sentence, we mentioned "could reflect the NK cell killing capacity and SIV DNA storage", because the number and proportion of NK cells are highly correlate with the effect of the killing capacity of NK cells; on the other hand, the higher the number or ratio of CD4+ T and HLADR- T cells, the higher their corresponding viral DNA load as viral reservoirs is likely to be. We therefore come up that "the ratio of NK cells versus CD4+ T and HLADR- T cells" could reflect to some extent the killing capacity of NK cells and the DNA stores of SIV. In the first paragraph of the Introduction, we have cited relevant literature to explain that resting CD4+ T cells act as potential viral reservoirs (closely related to intracellular viral load), and therefore have not repeated it here. Please see lines 50-58 in the revised manuscript.

  1. How did you calculate the P values in Table 1?

A: Thank you very much for your careful review. In Table 1, as the data do not obey a normal distribution, we have adopted the Friedman test (The Friedman non-parametric repeated measures ANOVA test) for the comparison of multiple groups of matched data (mentioned in line 207 in the methods). To make the presentation more comprehensive, we have added a description of the statistical analysis methods in the footnotes to Table 1. Please see lines 382-384 in the revised manuscript.

  1. Please explain why most of the statistics in figures had identical P values (multiples of 0.008)? For example, most two asterisks signified P=0.008, while one asterisk signified P=0.016, 0.024 or 0.032.

A: Thank you very much for bringing up the confusion. We would quite willingly explain that this is due to the fact that the data does not obey a normal distribution, so the statistical methods we have adopted were all non-parametric rank tests, which means that the statistical tests are based on calculating a rank sum and then ranking to obtain a P-value. This, coupled with the fact that our sample size is small, resulting in very limited ranking, makes it easy for duplicate p-values to occur, which is due to the nature of the statistical method itself and does not affect the results of the knotted statistics themselves.

Reviewer 3 Report

In the manuscript titled “Associations between NK cells in different immune organs and intracellular SIV DNA and RNA in regional resting CD4+ T cells in chronically SIVmac239-infected, treatment-naïve rhesus macaques” by Li et al, the authors describe NK cell phenotypes in blood and secondary lymphoid organs after chronic SIVmac239 infection (2 years) as well as assess cellular SIV RNA and DNA, amounts and ratios in the same tissues. The main findings of this paper are that cytotoxic NK cells as absent from the LN and that NK cell percentage is associated with SIV DNA/RNA ratios in resting memory T cells. Overall, the NK cell data is well presented, however this reviewer feels the leap made to conclude that NK cells are associate with and therefore maintain latency in the blood but not LNs is largely unsupported. The data as it stands only associated NK cells with DNA/RNA ratios in blood but provides no evidence that the presence of those cells has an effect on maintaining latency.

Major concerns

NK cell phenotyping/function (figure 1-4, supplemental)

1.     The viral load graph in supplemental argues some of the animals are not chronic progressors (ie don’t have an expected viral progression) at least compared to other studies that have used mac239 IR. Can the authors explain why the infections took 40-60 days to take hold and how this may have altered the NK cell data presented here?

2.     All of the macaques used in this paper were female, were the health donors also female? Additionally, why weren’t NKs assessed in spleen and LN from the healthy donors? It seems that having cytotoxic NK cells in the LN in general would be problematic even in health macaques and it would be important to understand if the lack of NK cells in the LN is due to SIV or just a normal immune response. Asother studies have published that NK cells are present at very low frequencies normally in LNs. If this data is not available then this should be addressed in the discussion. As there is no evidence that SIV results in limited NK cell infiltration into the LN.

3.     Was singlet gating completed initially before jumping to NK cell gating to eliminate any doublets?

4.     I believe that the wrong statistical test has been used throughout the manuscript. If multiple comparisons are being done within groups a more appropriate test would be a Kruskal Wallis test. This would still account for the nonparametric data however will take into account multiple comparisons.

5.     Figure 4 it is unclear if these were whole PBMCs/Spleen or isolated NKs in the NK functional assays

6.     Why only assess single positive cells and not look at combinations? (FACS data figure 3 and 4)

Resting T cell/latency (figure 5 and table 1)

1.     HLADR- is not enough to specifically isolate resting CD4s, typically cells are designated as DR-CD69-CD25-. What is the justification for using this single marker to identify these cells? Additionally, the authors keep referring to HLADR- T cells, are the referring to CD4 only? Please clarify throughout text.

2.     The data in table 1 make sense, DR- cells “resting” have more DNA per million, but equivalent RNA, which is what leads to a larger DNA/RNA ratio. However, the fact that the RNA levels are similar in both populations argues that a lot of the DR- population is not “latent” and the additional DNA signal may just eb defect genomes as the authors look at bulk DNA and not intact or integrated. Additionally, can CD4s be latent without ART?

3.     It is not clear how the P values are generated in table 1 – please explain what groups were compared and how.

4.     The claims from the NK% vs DNA/RNA ratios are not well supported and too strong. First, there is no evidence that similar numbers of NK cells are present in the LN in a healthy monkey and therefore have nothing to do with SIV control or lack thereof. Secondly, DR- CD4s in the periphery vs the LN are likely very different cells, and the correlation the authors are seeing may be due to compartmentalization. Finally, the correlations should be done on each compartment, not as one data set because the LN NK% are too low to assess and significantly skew the data set. One experiment that would be interesting and give more support to the claim that NK cells in the periphery drive CD4 latency would be to do an in vitro assay with CD4s from and SIV monkey +/- NK cells and to determine if the NKs are killing or driving the T cells towards latency.

5.     There is no control data available to support this statement and it should be removed (line 408): Considering extremely low NK cell counts, dominant non-cytotoxic subsets and exhausted phenotypes, the weaker immune pressure of LN-infiltrating NK cells may have limited effects on the reactivation of SIV reservoirs and viral replication.

6.     The conclusion statement is not supported, the data do not address reactivation of SIV in T cells and only fine a correlation between NK cell number and DNA/RNA in the blood, the statement needs significant toning down. Line 445  Overall, the results of our study indicated that NK cells could play a role in the inhibition of the reactivated SIV DNA in vivo. Given the limited number of NK cells in LNs, the influence of NK cells on the long-term accumulation of latent SIV DNA or RNA load in LNs could be constrained in the chronic viral infection process. The development of NK cell-directed treatment approaches aiming for HIV clearance remains challenging.

Author Response

Point-by-point responses to the comments from the Editors and Reviewers

Reviewer 3

Comments and Suggestions for Authors

In the manuscript titled “Associations between NK cells in different immune organs and intracellular SIV DNA and RNA in regional resting CD4+ T cells in chronically SIVmac239-infected, treatment-naïve rhesus macaques” by Li et al, the authors describe NK cell phenotypes in blood and secondary lymphoid organs after chronic SIVmac239 infection (2 years) as well as assess cellular SIV RNA and DNA, amounts and ratios in the same tissues. The main findings of this paper are that cytotoxic NK cells as absent from the LN and that NK cell percentage is associated with SIV DNA/RNA ratios in resting memory T cells. Overall, the NK cell data is well presented, however this reviewer feels the leap made to conclude that NK cells are associate with and therefore maintain latency in the blood but not LNs is largely unsupported. The data as it stands only associated NK cells with DNA/RNA ratios in blood but provides no evidence that the presence of those cells has an effect on maintaining latency.

Major concerns

NK cell phenotyping/function (figure 1-4, supplemental)

  1. The viral load graph in supplemental argues some of the animals are not chronic progressors (ie don’t have an expected viral progression) at least compared to other studies that have used mac239 IR. Can the authors explain why the infections took 40-60 days to take hold and how this may have altered the NK cell data presented here?

A: Thank you very much for raising this concern. In fact, the data in the supplementary Figure are long-term monitoring data after the SIV virus infected the macaques. Since we were intended to use the animal model of untreated chronic infection, the days of infection progression is not competitive with other studies. We adopted this chronic infection model for the purposes of observing NK cells and virus in different tissues during the stable phase of infection. In the graphs for long-term monitoring, we provide the results of monitoring throughout the course of the infection, but the formal experiments were performed after the acute phase of viral infection had passed and when the infection had stabilized. As to why the infection process takes 40-60 days to stabilize, this result is unfortunately outside the scope of our study and we have no clue.

  1. All of the macaques used in this paper were female, were the health donors also female? Additionally, why weren’t NKs assessed in spleen and LN from the healthy donors? It seems that having cytotoxic NK cells in the LN in general would be problematic even in health macaques and it would be important to understand if the lack of NK cells in the LN is due to SIV or just a normal immune response. Another studies have published that NK cells are present at very low frequencies normally in LNs. If this data is not available then this should be addressed in the discussion. As there is no evidence that SIV results in limited NK cell infiltration into the LN.

A: Thank you very much for your review comments. Indeed, in order to control against the infected group, the rhesus monkeys in the healthy group were matched to the infected group for both gender and age. To complete the content, we have added a description in the methods section. Please see line 99 in the revised manuscript. However, we would like to clarify that, given the extremely precious sample of healthy rhesus monkeys, only peripheral blood from healthy donors was taken for control purposes and it was not possible to execute the animals to obtain the corresponding single cell suspensions of spleen and LN. Aware of this shortcoming, in this manuscript, the frequency and function of NK cells in PBMC of all healthy rhesus macaques were strictly compared to those of infected rhesus macaques alone, rather than to those of infected LN and spleen. We also highly endorse that NK cells in LN are inherently low and not caused by SIV. Therefore, we have mentioned in our discussion that "LN-infiltrated NK cells could have a limited impact on SIV viral activity, given the extremely low NK cell numbers, limited cytotoxic subpopulations, and depleted phenotype". Please see lines 425-428 in the revised manuscript. (This sentence has been deleted according to the suggestions)

  1. Was singlet gating completed initially before jumping to NK cell gating to eliminate any doublets?

A: Thank you very much for your question. Before performing NK cell gating, we first completed singlet gating for flow cytometric analysis, which included FSC/SSC gating, live cell gating, and removal of adherent cells. However, as these were not the core results of the experiments, they are not shown in detail in the figure. To clarify the gating strategy of NK cells, we have added a description in the manuscript. Please see lines 222-225 in the revised manuscript.

  1. I believe that the wrong statistical test has been used throughout the manuscript. If multiple comparisons are being done within groups a more appropriate test would be a Kruskal Wallis test. This would still account for the nonparametric data however will take into account multiple comparisons.

A: Thank you very much for your instructive suggestion. We have previously considered using a non-parametric Kruskal Wallis test for comparing differences between groups. However, given that this statistical method can only give an overall assessment statistical difference, it cannot provide a specific result exactly where the difference is. Therefore, we adopted the more appropriate Mann Whitney U test after careful consideration, which could present a more visual picture of which two groups the difference exists between by comparing groups in succession. Moreover, in Table 1, we have compared the overall differences between multiple groups. But given that the data do not follow a normal distribution and that there is a paired relationship between the data, we have taken the corresponding Friedman test, which should be consistent with your suggestion.

  1. Figure 4 it is unclear if these were whole PBMCs/Spleen or isolated NKs in the NK functional assays

A: Thank you very much for raising this concern. In Figure 4, we have isolated "lymphocytes" from peripheral blood and spleen single-cell suspensions for the experiment, and tested the function of the NK cell subsets by flow cytometry gating. As the flow assay method has been mentioned in the methods section, and the NK cells gate process is the same as previously described, it has not been repeated here. For clarity of meaning, we have added a description of the content in the figure legend. Please see lines 372-377 in the revised manuscript.

  1. Why only assess single positive cells and not look at combinations? (FACS data figure 3 and 4)

A: Thank you for your instructive suggestions. It is true that different combinations of results can be presented in the FACS data presentation, but confusion can also arise. Combinations could complicate the presentation of results, and the manner and choice of combination may be subjective, resulting in a less specific presentation of results that leaves out details of single positive cells. We ultimately decided to present single positive cells separately, thus ensuring the results are as objective as possible.

Resting T cell/latency (figure 5 and table 1)

  1. HLADR- is not enough to specifically isolate resting CD4s, typically cells are designated as DR-CD69-CD25-. What is the justification for using this single marker to identify these cells? Additionally, the authors keep referring to HLADR- T cells, are the referring to CD4 only? Please clarify throughout text.

A: Thank you very much for your instructive advice. You are absolutely right that CD3+CD4+CD8-CD25-CD69-HLA-DR-is indeed the most accurate flow metric for sorting resting CD4+ T cells currently. However, considering that such fine gating requires a large number of samples, resulting in insufficient remaining cells to support subsequent experiments, this greatly increases the difficulty of flow experiments, we used HLADR-sorting circle gate to observe the activation of T cells on the basis of T cell gate. In light of your suggestion, we are aware of the inappropriateness of claims for "resting CD4+ T cells" in the manuscript and have therefore revised the relevant description throughout the text (CD3+ CD4+ CD8- HLADR- T cells/ HLADR- CD4+ T cells) to ensure accuracy. Please see lines 191-311-327 in the revised manuscript. In addition, we have been referring to HLADR-T cells, specifically HLADR-cells among CD4+ T cells. For clarification, we have refined this description throughout the manuscript. Please see lines 311-327 and the new Table 1 in the revised manuscript.

  1. The data in table 1 make sense, DR- cells “resting” have more DNA per million, but equivalent RNA, which is what leads to a larger DNA/RNA ratio. However, the fact that the RNA levels are similar in both populations argues that a lot of the DR- population is not “latent” and the additional DNA signal may just eb defect genomes as the authors look at bulk DNA and not intact or integrated. Additionally, can CD4s be latent without ART?

A: Thank you very much for raising this concern. It is a pity that the DNA assay we used could not identify whether the DNA is integrated or defective. In fact, we would like to measure the cellular DNA and RNA levels as a whole to reflect, to some extent, the DNA/RNA levels. We have to state that this is not representative of the viral reservoir load and we were particularly careful not to mention the concept of "viral reservoir" throughout the text because we are fully aware that these animals have not been treated with ART. The term "latent" is intended to be distinguished from "reservoir" and indicating that the intracellular virus could be transcribed actively, or released at any time in this state. If there is a better alternative, we would be pleased to modify it.

  1. It is not clear how the P values are generated in table 1 – please explain what groups were compared and how.

A: Thank you very much for your careful review. In Table 1, as the data do not obey a normal distribution, we have adopted the Friedman test (The Friedman non-parametric repeated measures ANOVA test) for the comparison of multiple groups of matched data (mentioned in line 210 in the methods). To make the presentation more complete, we have added a description of the statistical analysis methods in the footnotes to Table 1. Please see lines 382-384 in the revised manuscript.

  1. The claims from the NK% vs DNA/RNA ratios are not well supported and too strong. First, there is no evidence that similar numbers of NK cells are present in the LN in a healthy monkey and therefore have nothing to do with SIV control or lack thereof. Secondly, DR- CD4s in the periphery vs the LN are likely very different cells, and the correlation the authors are seeing may be due to compartmentalization. Finally, the correlations should be done on each compartment, not as one data set because the LN NK% are too low to assess and significantly skew the data set. One experiment that would be interesting and give more support to the claim that NK cells in the periphery drive CD4 latency would be to do an in vitro assay with CD4s from and SIV monkey +/- NK cells and to determine if the NKs are killing or driving the T cells towards latency.

A: Thank you very much for your suggestion. As healthy rhesus macaque samples are very precious and cannot be executed, we regret that we cannot provide evidence related to LN in healthy monkeys. Regarding the point that assessing the correlation separately for each compartment, we have tried this before, but the sample size of each tissue and organ is too small to present a significant correlation when compared separately, so we can only assess the correlation in the overall context. In addition, we strongly acknowledge your comment about whether NK cells kill or drive CD4+T cells latency. Further experiments are required subsequently to provide supporting evidence. Therefore, to prevent overstatement, we have revised the conclusion and presentation of this paragraph. Please see lines 329-332 and lines 429-432 in the revised manuscript.

  1. There is no control data available to support this statement and it should be removed (line 408): Considering extremely low NK cell counts, dominant non-cytotoxic subsets and exhausted phenotypes, the weaker immune pressure of LN-infiltrating NK cells may have limited effects on the reactivation of SIV reservoirs and viral replication.

A: Thank you very much for your advice. For more accurate representation, we have removed this sentence based on your comment guidance. Please see lines 425-428 in the revised manuscript.

  1. The conclusion statement is not supported, the data do not address reactivation of SIV in T cells and only fine a correlation between NK cell number and DNA/RNA in the blood, the statement needs significant toning down. Line 445 Overall, the results of our study indicated that NK cells could play a role in the inhibition of the reactivated SIV DNA in vivo. Given the limited number of NK cells in LNs, the influence of NK cells on the long-term accumulation of latent SIV DNA or RNA load in LNs could be constrained in the chronic viral infection process. The development of NK cell-directed treatment approaches aiming for HIV clearance remains challenging.、

A: Thank you very much for your insightful comments. It was slightly abrupt for us to jump to this hasty conclusion. We have been acutely aware that our data are more informative in terms of NK cell frequency and SIV DNA/RNA correlation. To make the conclusions more objective and reasonable, we have rewritten the sentence and narrowed its content. Please see lines 41-46 and 457-461 in the revised manuscript.

Round 2

Reviewer 3 Report

This reviewer thanks the authors for careful revision of the manuscript titled “Associations between NK cells in different immune organs and cellular SIV DNA and RNA in regional HLADR- CD4+ T cells in chronically SIVmac239-infected, treatment-naïve rhesus macaques”. There are still a few points that require further revision or clarification.

1. The viral load graph in supplemental argues some of the animals are not chronic progressors (ie don’t have an expected viral progression) at least compared to other studies that have used mac239 IR. Can the authors explain why the infections took 40- 60 days to take hold and how this may have altered the NK cell data presented here?

A: Thank you very much for raising this concern. In fact, the data in the supplementary Figure are long-term monitoring data after the SIV virus infected the macaques. Since we were intended to use the animal model of untreated chronic infection, the days of infection progression is not competitive with other studies. We adopted this chronic infection model for the purposes of observing NK cells and virus in different tissues during the stable phase of infection. In the graphs for long-term monitoring, we provide the results of monitoring throughout the course of the infection, but the formal experiments were performed after the acute phase of viral infection had passed and when the infection had stabilized. As to why the infection process takes 40-60 days to stabilize, this result is unfortunately outside the scope of our study and we have no clue.

R3: Thank you for the thorough explanation. In an effort for clarity, please add this information to the results and indicated that NK cell numbers and function were assessed only once the infection was stable.

 4. I believe that the wrong statistical test has been used throughout the manuscript. If multiple comparisons are being done within groups a more appropriate test would be a Kruskal Wallis test. This would still account for the nonparametric data however will take into account multiple comparisons.

A: Thank you very much for your instructive suggestion. We have previously considered using a non-parametric Kruskal Wallis test for comparing differences between groups. However, given that this statistical method can only give an overall assessment statistical difference, it cannot provide a specific result exactly where the difference is. Therefore, we adopted the more appropriate Mann Whitney U test after careful consideration, which could present a more visual picture of which two groups the difference exists between by comparing groups in succession. Moreover, in Table 1, we have compared the overall differences between multiple groups. But given that the data do not follow a normal distribution and that there is a paired relationship between the data, we have taken the corresponding Friedman test, which should be consistent with your suggestion.

R3: The Mann Whitney U test cannot be used across multiple groups as is being done in Figs 1-3 because it overestimates the significance. The Kruskal Wallis test (or ANOVA if parametric) needs to be used and can be paired with Dunn’s correction to obtain specific P values with the comparisons. This can easily be done in graphpad. In figures 1-3 there are clear multiple comparisons between the SIV conditions, and those p values need to be adjusted. Using Mann Whitney U in Figure 4 seems fine since the authors are only comparing HD PBMC to SIV PBMC or SIV PBMC to SIV spleen and they appear to be different questions.   

https://www.graphpad.com/guides/prism/latest/statistics/stat_nonparametric_tests_dont_compa.htm

https://www.graphpad.com/guides/prism/latest/statistics/how_the_kruskal-wallis_test_works.htm

9. It is not clear how the P values are generated in table 1 – please explain what groups were compared and how.

A: Thank you very much for your careful review. In Table 1, as the data do not obey a normal distribution, we have adopted the Friedman test (The Friedman non-parametric repeated measures ANOVA test) for the comparison of multiple groups of matched data (mentioned in line 210 in the methods). To make the presentation more complete, we have added a description of the statistical analysis methods in the footnotes to Table 1. Please see lines 382-384 in the revised manuscript.

R3: This is still not clear to me. Can you please indicate in the methods or results the exact groups that are compared to obtain the P values? I agree with the test bring used, but it is unclear to me the comparison done to obtain this result. For example, when you say “we have compared the overall differences between multiple groups” how exactly is that being done.

12. The conclusion statement is not supported, the data do not address reactivation of SIV in T cells and only fine a correlation between NK cell number and DNA/RNA in the blood, the statement needs significant toning down. Line 445 Overall, the results of our study indicated that NK cells could play a role in the inhibition of the reactivated SIV DNA in vivo. Given the limited number of NK cells in LNs, the influence of NK cells on the long-term accumulation of latent SIV DNA or RNA load in LNs could be constrained in the chronic viral infection process. The development of NK cell-directed treatment approaches aiming for HIV clearance remains challenging.

A: Thank you very much for your insightful comments. It was slightly abrupt for us to jump to this hasty conclusion. We have been acutely aware that our data are more informative in terms of NK cell frequency and SIV DNA/RNA correlation. To make the conclusions more objective and reasonable, we have rewritten the sentence and narrowed its content. Please see lines 41-46 and 457-461 in the revised manuscript.

R3: Thank you for the careful revision of this statement. To further clarify please alter the language to state “Given the scarcity of NK cells in LNs, the cytotoxicity effect of NK cells on the SIV replication in LNs is likely limited during chronic viral infection.” LINE 460-461

Author Response

Point-by-point responses to the comments from the Editors and Reviewers

Reviewer 3

Comments and Suggestions for Authors

This reviewer thanks the authors for careful revision of the manuscript titled “Associations between NK cells in different immune organs and cellular SIV DNA and RNA in regional HLADR- CD4+ T cells in chronically SIVmac239-infected, treatment-naïve rhesus macaques”. There are still a few points that require further revision or clarification.

  1. The viral load graph in supplemental argues some of the animals are not chronic progressors (ie don’t have an expected viral progression) at least compared to other studies that have used mac239 IR. Can the authors explain why the infections took 40- 60 days to take hold and how this may have altered the NK cell data presented here?

A: Thank you very much for raising this concern. In fact, the data in the supplementary Figure are long-term monitoring data after the SIV virus infected the macaques. Since we were intended to use the animal model of untreated chronic infection, the days of infection progression is not competitive with other studies. We adopted this chronic infection model for the purposes of observing NK cells and virus in different tissues during the stable phase of infection. In the graphs for long-term monitoring, we provide the results of monitoring throughout the course of the infection, but the formal experiments were performed after the acute phase of viral infection had passed and when the infection had stabilized. As to why the infection process takes 40-60 days to stabilize, this result is unfortunately outside the scope of our study and we have no clue.

R3: Thank you for the thorough explanation. In an effort for clarity, please add this information to the results and indicated that NK cell numbers and function were assessed only once the infection was stable.

A: Many thanks to the reviewers for the suggestion. Based on this suggestion, we have added this information to the methods and results, specifically stating that all experiments were started to assess NK cell numbers and function only after the infection was stable. Please see the lines 96-97 and lines 220-221 in the revised manuscript.

  1. I believe that the wrong statistical test has been used throughout the manuscript. If multiple comparisons are being done within groups a more appropriate test would be a Kruskal Wallis test. This would still account for the nonparametric data however will take into account multiple comparisons.

A: Thank you very much for your instructive suggestion. We have previously considered using a non-parametric Kruskal Wallis test for comparing differences between groups. However, given that this statistical method can only give an overall assessment statistical difference, it cannot provide a specific result exactly where the difference is. Therefore, we adopted the more appropriate Mann Whitney U test after careful consideration, which could present a more visual picture of which two groups the difference exists between by comparing groups in succession. Moreover, in Table 1, we have compared the overall differences between multiple groups. But given that the data do not follow a normal distribution and that there is a paired relationship between the data, we have taken the corresponding Friedman test, which should be consistent with your suggestion.

R3: The Mann Whitney U test cannot be used across multiple groups as is being done in Figs 1-3 because it overestimates the significance. The Kruskal Wallis test (or ANOVA if parametric) needs to be used and can be paired with Dunn’s correction to obtain specific P values with the comparisons. This can easily be done in graphpad. In figures 1-3 there are clear multiple comparisons between the SIV conditions, and those p values need to be adjusted. Using Mann Whitney U in Figure 4 seems fine since the authors are only comparing HD PBMC to SIV PBMC or SIV PBMC to SIV spleen and they appear to be different questions.  

A: Thank you very much for your insightful suggestion. We have discussed this issue in depth and consulted with statistical experts in order to refine the statistical methods and obtain reliable results. According to the advice of the statistical experts, after confirming that the data do not conform to a normal distribution, we should first assess the multiple group differences using the Kruskal Wallis test. Next, the specific P-values between the two groups can be obtained using Dunn post hoc tests or Mann Whitney U test with the assurance that there is a difference between multiple groups. Therefore, we supplemented the results of the Kruskal Wallis test with the corresponding explanations in the text (since data form healthy rhesus peripheral blood cannot be directly compared with that form infected rhesus’ lymphoid organs such as the spleen, we only assessed the overall differences among the SIV-infected group). Please see the lines 237-294 and lines 349-386 of the revised manuscript. Additionally, it is important to explain that we did not actually use the Mann Whitney U test for multiple comparisons, but only for comparisons between two groups. Realizing that the original graph could be misleading, we further refined the graph presentation to distinguish between multiple comparisons and P-values obtained from two-group comparisons. Please see the revised Figure 1- Figure 3.

  1. It is not clear how the P values are generated in table 1 – please explain what groups were compared and how.

A: Thank you very much for your careful review. In Table 1, as the data do not obey a normal distribution, we have adopted the Friedman test (The Friedman non-parametric repeated measures ANOVA test) for the comparison of multiple groups of matched data (mentioned in line 210 in the methods). To make the presentation more complete, we have added a description of the statistical analysis methods in the footnotes to Table 1. Please see lines 382-384 in the revised manuscript.

R3: This is still not clear to me. Can you please indicate in the methods or results the exact groups that are compared to obtain the P values? I agree with the test bring used, but it is unclear to me the comparison done to obtain this result. For example, when you say “we have compared the overall differences between multiple groups” how exactly is that being done.

A: Thank you very much for raising this concern, and we are more than pleased to explain it. In Table 1, we have compared the differences in the overall distribution of data from the four groups of PBMCs, PaLNs, ALNs and Spleen after SIV infection. As shown in the table, each animal corresponds to a number, so in practice, we wish to compare samples of different tissue or organ origin from the same animal. Considering this pairwise relationship, we therefore chose the Friedman test, which is suitable for testing non-parametric multiple paired samples, to assess multiple differences between groups (the overall differences we mentioned). This statistical procedure was also implemented through the Graphpad software. To clarify this, we specified the exact groups for comparison in the footnotes of Table 1. Please refer to the lines 401-403 in the revised manuscript for details.

  1. The conclusion statement is not supported, the data do not address reactivation of SIV in T cells and only fine a correlation between NK cell number and DNA/RNA in the blood, the statement needs significant toning down. Line 445 Overall, the results of our study indicated that NK cells could play a role in the inhibition of the reactivated SIV DNA in vivo. Given the limited number of NK cells in LNs, the influence of NK cells on the long-term accumulation of latent SIV DNA or RNA load in LNs could be constrained in the chronic viral infection process. The development of NK cell-directed treatment approaches aiming for HIV clearance remains challenging.

A: Thank you very much for your insightful comments. It was slightly abrupt for us to jump to this hasty conclusion. We have been acutely aware that our data are more informative in terms of NK cell frequency and SIV DNA/RNA correlation. To make the conclusions more objective and reasonable, we have rewritten the sentence and narrowed its content. Please see lines 41-46 and 457-461 in the revised manuscript.

R3: Thank you for the careful revision of this statement. To further clarify please alter the language to state “Given the scarcity of NK cells in LNs, the cytotoxicity effect of NK cells on the SIV replication in LNs is likely limited during chronic viral infection.” LINE 460-461

A: Thank you very much for your patient guidance. In the updated manuscript, we have modified the sentence as you suggested to clarify the statement. Please see lines 476-478 in the revised manuscript for more details.
